# Stratospheric influence on surface ozone pollution in China

Zhixiong Chen[1,2], Jane Liu [1,3] ✉, Xiushu Qie [2] ✉, Xugeng Cheng[1], Mengmiao Yang[1], Lei Shu [1] & Zhou Zang[3]

Events of stratospheric intrusions to the surface (SITS) can lead to severe ozone ($O_3$) pollution. Still, to what extent SITS events impact surface $O_3$ on a national scale over years remains a long-lasting question, mainly due to difficulty of resolving three key SITS metrics: frequency, duration and intensity. Here, we identify 27,616 SITS events over China during 2015-2022 based on spatiotemporally dense surface measurements of $O_3$ and carbon monoxide, two effective indicators of SITS. An overview of the three metrics is presented, illustrating large influences of SITS on surface $O_3$ in China. We find that SITS events occur preferentially in high-elevation regions, while those in plain regions are more intense. SITS enhances surface $O_3$ by 20 ppbv on average, contributing to 30-45% of $O_3$ during SITS periods. Nationally, SITS-induced $O_3$ peaks in spring and autumn, while over 70% of SITS events during the warm months exacerbate $O_3$ pollution. Over 2015-2022, SITS-induced $O_3$ shows a declining trend. Our observation-based results can have implications for $O_3$ mitigation policies in short and long terms.

Compelling evidence has suggested that high ozone ($O_3$) concentrations in the surface layer can be harmful to human health and vegetation growth[1]. As known, tropospheric $O_3$ originates from two sources: photochemical production within the troposphere and dynamical injection from the stratosphere. Though the injected stratospheric $O_3$ is estimated to account for only 5-10% of the tropospheric $O_3$ sources[2], case studies of stratospheric intrusions (SI) to the troposphere have documented how SI occurred and impacted tropospheric and even surface $O_3$[3-5]. Events of deep and fast stratospheric intrusions to the surface (SITS) can lead to high-$O_3$ episodes, inducing severe $O_3$ pollution[6-9]. In the recent decade, China has confronted a severe $O_3$ pollution problem, even after the implementation of strict policies on emission reductions of $O_3$ precursors. The causes for this environmental issue have been investigated from the perspectives of chemical responses to the changes in both emissions and meteorology[10-12]. Yet, the contribution of natural stratospheric $O_3$ inputs to surface $O_3$ pollution is often neglected. Verstraeten et al.[13] found a substantial positive trend in stratospheric contributions to tropospheric $O_3$ over China

for 2005-2010 based on numerical simulations. It is still highly uncertain about the influence of SITS on the surface $O_3$ over a long period. Such long-term variation of stratospheric influence on surface $O_3$ is not only an issue for China, but also for elsewhere in the world[3].

Though SITS events are transient and limited in local areas, their impact on surface $O_3$ can be substantial during SITS periods over the affected areas, regarding the absolute $O_3$ enhancement and fractional contribution to overall surface $O_3$. For example, Chen et al.[14] showed that a SITS event during the COVID-19 lockdown period in 2020 enhanced surface $O_3$ concentrations in Beijing, China, by 8 ppbv, which contributed to 23% of overall surface $O_3$. These extra stratospheric $O_3$ inputs, plus the background $O_3$, can exacerbate air pollution beyond the recommended $O_3$ threshold. The SITS-induced $O_3$ exceedances depend not only on the absolute SITS $O_3$ inputs, but also on surface background $O_3$, which is modulated by the interplay of multiple chemical and physical processes, varying at different time scales, and sensitive to $O_3$ precursor emissions and environmental conditions. Therefore, to what extent SITS events are harmful to

[1]Key Laboratory for Humid Subtropical Eco-Geographical Processes of the Ministry of Education, School of Geographical Sciences, Fujian Normal University, Fuzhou, China. [2]Institute of Atmospheric Physics, Chinese Academy of Sciences, Beijing, China. [3]Department of Geography and Planning, University of Toronto, Toronto, ON, Canada. ✉e-mail: janejj.liu@utoronto.ca; qiex@mail.iap.ac.cn

human health and crop yield is largely uncertain, not only in their absolute magnitudes, but also in relative contributions to surface $O_3$. To our knowledge, this issue has not been resolved on a national scale for the worsening ground-level $O_3$ pollution in China to date.

Assessing the impact of SITS on surface $O_3$ for large areas over long periods requires a good understanding of three key SITS metrics: frequency, duration, and intensity. These three metrics are essential to evaluate the $O_3$ exposure associated with SITS concerning the health effect. Despite the knowledge obtained from SITS case studies, it is challenging to obtain an overview of SITS events in terms of these three metrics on a national, continent, or global scale, due to the difficulty of explicitly resolving them. For stratospheric air to reach the surface, it has to be transported downward across the first barrier, the tropopause, through deep SI, and then quickly descends and crosses the second barrier, the planetary boundary layer (PBL)[15–17]. Insufficient consideration of multi-scale atmospheric dynamical processes, including large-scale tropopause folding and small-scale PBL mixing, can easily lead to biased representation of the frequency, duration, and intensity of SITS events. Chemical transport models are often applied to evaluate stratospheric $O_3$ transported into the troposphere using tagged stratospheric tracers[18,19], but the estimated stratospheric influences are heavily dependent on tracer definitions and resolvable dynamical processes[20,21]. Especially, the abundant and complicated chemical sink compounds of $O_3$ in the PBL would result in enhanced uncertainty of the intruded stratospheric $O_3$ if they are not accurately described in models[7,16]. The instantaneous and local nature of SITS events[3] requires comprehensive observations with a high spatial and temporal resolution in order to assess their impacts on surface $O_3$. The routine weekly ozonesonde observations can directly monitor vertical $O_3$ variation, they are not able to capture SITS events that persist only for several hours[4,8,9]. Running multi-instrumental campaigns with targeted stratospheric tracers, such as $O_3$ and cosmogenic radionuclide, can provide reliable indicators of SITS[5,22,23]. Performed in limited regions and periods, however, these campaigns are spatially and temporally too scant to provide a national-scale estimation of stratospheric influences on surface $O_3$ for long periods. Therefore, how can we overcome our insufficient ability to achieve a nation-wide assessment of SITS events regarding their frequency, duration, and intensity?

Stratospheric air is characterized by high $O_3$, low carbon monoxide (CO), and low relative humidity (RH)[3,4,24]. Intrusions of stratospheric air downward to the surface level produce sharp upward (downward) spikes in the surface-measured $O_3$ (CO), as documented in many case studies[6–9]. Such stratospheric signals can provide a clear indication of SI that have reached the surface level with the aid of dense surface observations. China has built a nationwide network consisting of more than 1600 surface stations that are capable of providing hourly observations of $O_3$, CO, $NO_2$ and $SO_2$ concentrations[25,26]. This comprehensive dataset is both spatially and temporally dense, so it is possible that the data from this network could capture the signals of SITS events over large areas much more effectively than those mentioned traditional data, and hence provide an opportunity to investigate the stratospheric contribution to surface $O_3$ and its long-term trends across the nation.

Here, we take advantage of the dense surface observations and develop a SITS detection method using $O_3$ and CO as stratospheric indicators and their distinct spikes during SITS as constraints (see "Methods" section), based on our previous case studies[9,14]. This method bypasses the need for detailed knowledge about the multi-scale dynamical processes and complicated chemical sinks for the descending stratospheric air. The general features of SITS from our detection method are in line with those derived from multi-instrumental observations in published literatures (see "Methods" section). This gives us confidence to resolve the spatial and temporal variations of SITS events in China. Basing on large samples of SITS events detected locally at individual stations across the nation in 8

years (27,616 events in total), here we aim to address the following scientific questions: (1) what are the characteristics of SITS events in China over 2015-2022 in terms of frequency, duration, and intensity? (2) to what extent SITS-induced $O_3$ impacts surface $O_3$ variations and contributes to the occurrences of surface $O_3$ pollution? and (3) what is the trend of SITS-induced $O_3$ in China over 2015-2022?

## Results

### Spatial distributions and seasonal variations of SITS in China

A total of 27,616 SITS events were screened out during 2015-2022 using surface observations from 1500 stations with continuous 8-year measurements in China (see "Methods" section). Figure 1 shows the spatial distributions of the annual frequency, annual total duration, and surface $O_3$ enhancements of the SITS averaged over 2015-2022. The hot spots for SITS frequency and duration are located in high-elevation regions in the southeast rim of the Tibetan Plateau in southwest China, Tianshan Mountains in northwest China and Changbai Mountains in northeast China (Fig. 1a, b). The annual SITS frequency averaged is 8-12 occurrences per year over these regions, with an average of the total duration of 120-150 h per year, i.e., ~1.7% of the entire year. The highest SITS frequency is 16 and 14 occurrences per year in cities Panzhihua (3130 m above sea level) and Chuxiong (2530 m above sea level), both of which are situated closely to the Tibetan Plateau. Correspondingly, the total duration of SITS there reaches 345 hours and 318 hours per year, respectively, accounting for 3.94% and 3.63% of the time in a year. In the plain areas of eastern China, however, SITS events are relatively sparse and rare with a mean frequency of 2–3 occurrences per year and a total duration of 30-60 hours annually.

Figure 1c shows the SITS-induced $O_3$ enhancements above their corresponding reference baselines averaged over the SITS duration (see "Methods" section for the baseline definition). Unlike the spatial variations in frequency and duration, the surface $O_3$ enhancements are larger in plain than in high-elevation areas. In central and eastern China, average surface $O_3$ enhancements are 15–25 ppbv during the SITS, while in western China, where elevations are high, only 7–15 ppbv. The SITS-induced $O_3$ enhancements varying with elevation are more evident in the Sichuan basin (SCB). The reasons for such $O_3$ enhancements varying with elevation can be subject to further studies. Here we suggest that for those SITS events descending to plain regions, the air parcels originated in the stratosphere have to travel a deeper vertical extension experiencing more chance to be diluted by tropospheric air. Hence, only those strong SITS events are more likely to reach the surface over plain regions. Trickl et al.[27] suggested that the intensity of O3 enhancements during the SITS partially depend on how high the intrusions start in the stratosphere. Possibly, SITS events over plain regions in eastern China may initiate at higher altitudes within the stratosphere.

The frequency of the SITS exhibits distinct seasonality with a maximum in early spring, a secondary maximum in autumn and a minimum in summer (Fig. 2a). Such seasonality is in agreement with multiple studies of SI climatology in the midlatitudes of the Northern Hemisphere[28–32]. In particular, the number of detected SITS events in China shows a pronounced peak in March and a minimum in August. In terms of the duration of the SITS events, responding to the rapid mixing processes with tropospheric air and abundant chemical sinks, SITS-induced $O_3$ can elevate surface $O_3$ concentrations above the baseline for 18 hours on average (Fig. 2b). In general, SITS occurring in spring and autumn persist longer than in winter and summer. Besides showing the highest SITS frequency, March also exhibits the longest duration of SITS events, yielding elongated stratospheric impacts on surface $O_3$. Note that the duration here does not include the time before which stratospheric $O_3$ is chemically destroyed in the troposphere, instead, it refers to the period when the stratospheric air parcel reaches the surface and retains its rich-$O_3$ and poor-CO properties that are distinguishable from tropospheric air.

                                                                          2

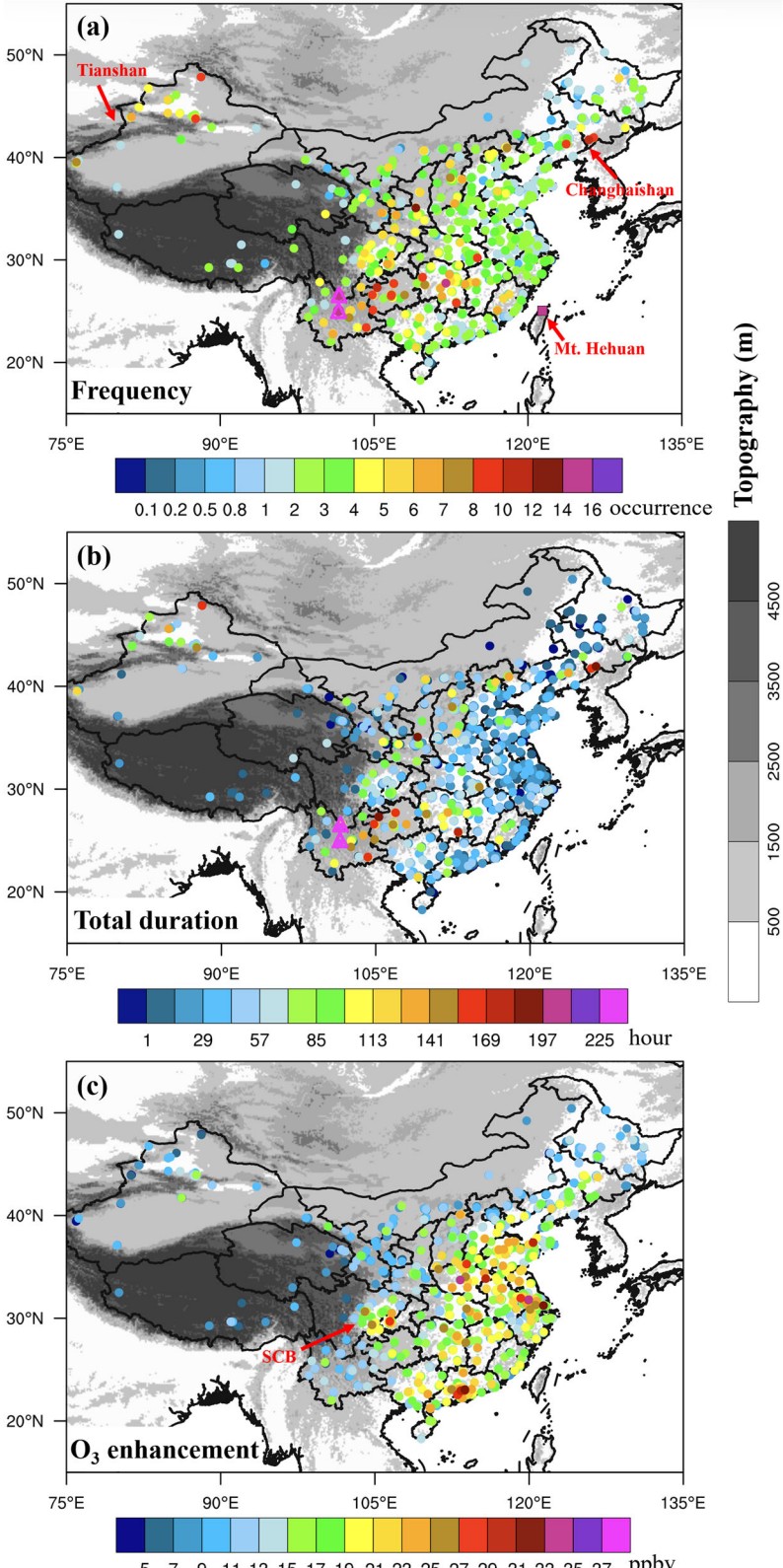

**Fig. 1 | Spatial distributions of the three key metrics of stratospheric intrusions to the surface (SITS) events in China. a** The annual frequency (unit: occurrence per year), (**b**) the annual total duration (unit: hours per year) and (**c**) the surface O₃ enhancements (unit: ppbv) of SITS events averaged over 2015-2022. Mt. Changbaishan, Tianshan and Hehuan are indicated in (**a**), and locations of Panzhihua and Chuxiong are marked by the magenta triangles. SCB in (**c**) refers to the Sichuan Basin. Source data are provided as a Source Data file.

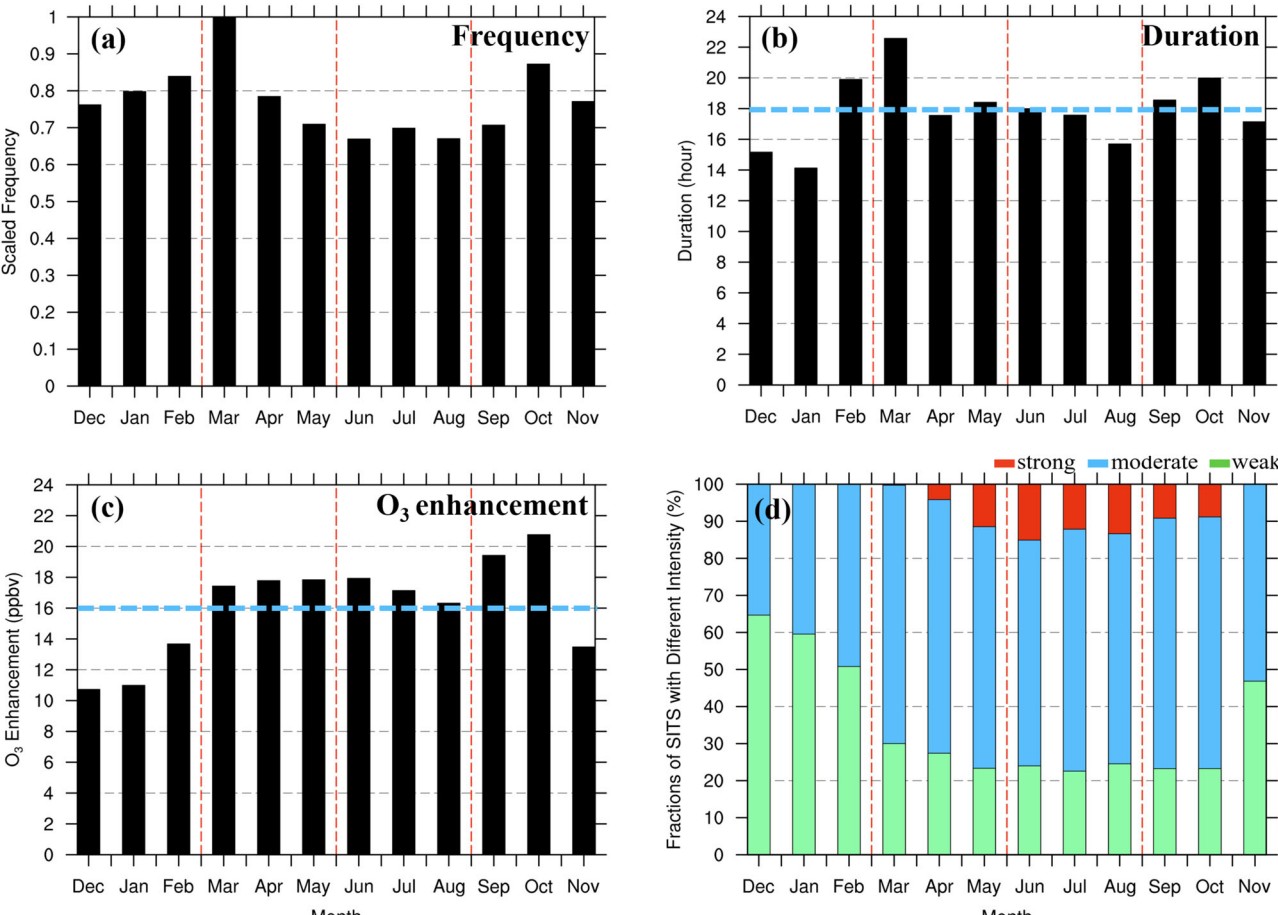

**Fig. 2 | Monthly variations in the three key metrics of stratospheric intrusions to the surface (SITS) events in China. a** Frequency of SITS events scaled to the maximum value in March. **b** Duration of STIS events in each month (unit: hours). **c** O₃ enhancements of SITS events relative to the corresponding surface O₃ reference baselines (unit: ppbv). The horizontal blue dashed lines in **b** and **c** represent the corresponding annual means. **d** Fractions of SITS associated with different intensity categories, i.e., weak (O₃ enhancements<15 ppbv, green bar), moderate (15 ppbv<O₃ enhancements<40 ppbv, blue bar), and strong (O₃ enhancements >40 ppbv, red bar). The four seasons are separated by red dashed lines for winter (December-February), spring (March-May), summer (June-August), and autumn (September-November). Source data are provided as a Source Data file.

The SITS-induced O₃ enhancements averaged over the SITS duration in each month are compared in Fig. 2c. Generally, the stratospheric O₃ enhancements are stronger in warmer months, with an average of 18.1 ppbv above the baseline value from March to October, and weaker during colder months, with an average of 12.2 ppbv above the baseline. Furthermore, Fig. 2d illustrates the distinct seasonal cycles of the SITS events with three levels of intensity measured by O₃ enhancements[27]. The SITS events with O₃ enhancements exceeding the surface O₃ baseline by less than 15 ppbv (weak SITS) exhibit a maximum in cold months, while those with an exceedance over 40 ppbv (strong SITS) appear frequently in warm months, especially in summer. The distinct seasonality of SITS with different intensities is in line with Trickl et al.[27], who attributed the maximum of stratospheric O₃ intensity in summer to the air parcels' higher origins in the stratosphere than in the other seasons. Due to an elevated tropopause in summer[33], the summertime SITS events originating at higher altitudes would have high O₃ concentrations because the altitudes are closer to the stratospheric O₃ reservoir ~20–25 km.

**Substantial stratospheric contribution to surface ozone pollution episodes**

Based on a complete depiction of the three key SITS metrics, we estimate the magnitude of injected stratospheric O₃ at ground level by integrating the hourly excess of O₃ concentrations relative to their reference baselines[28] (Fig. 3a, unit: ppbv*hour, see "Methods" section). Averaged over 2015-2022, the monthly stratospheric O₃ inputs to the surface layer in China show a peak in March, a second peak in October and a minimum in December and January. Governed by the three key metrics of SITS, it is clear that stratospheric influences maximize in early spring due to the high frequency and duration and moderate intrusion intensity. The combination of high intrusion intensity and moderate frequency and duration of SITS in autumn leads to a second peak of stratospheric influences. The intrusion frequency is relatively high in winter (Fig. 2a); however, the short duration and weak magnitudes of the SITS (Fig. 2b, c) result in the least stratospheric inputs to surface O₃ in winter. During the short periods of the SITS (referred to as the SITS duration), these additional stratospheric O₃ inputs substantially enhance the surface O₃ concentrations, consisting of 30-45% of surface O₃ over SITS-affected areas (Fig. 3b–e). The ratio of stratospheric O₃ to overall surface O₃ can reach 58% in March and October, calling for extra consideration of the stratospheric influence on O₃ budgets in the 2 months.

The natural stratosphere-to-troposphere (STT) processes have direct influences on the chemical compositions in the troposphere[34]. Figure 4 presents a statistical overview of the instantaneous impacts of SITS on surface gaseous compounds, from 12 h before the start of the SITS to 20 hours after, averaged over all the detected SITS events. The synchronously sharp

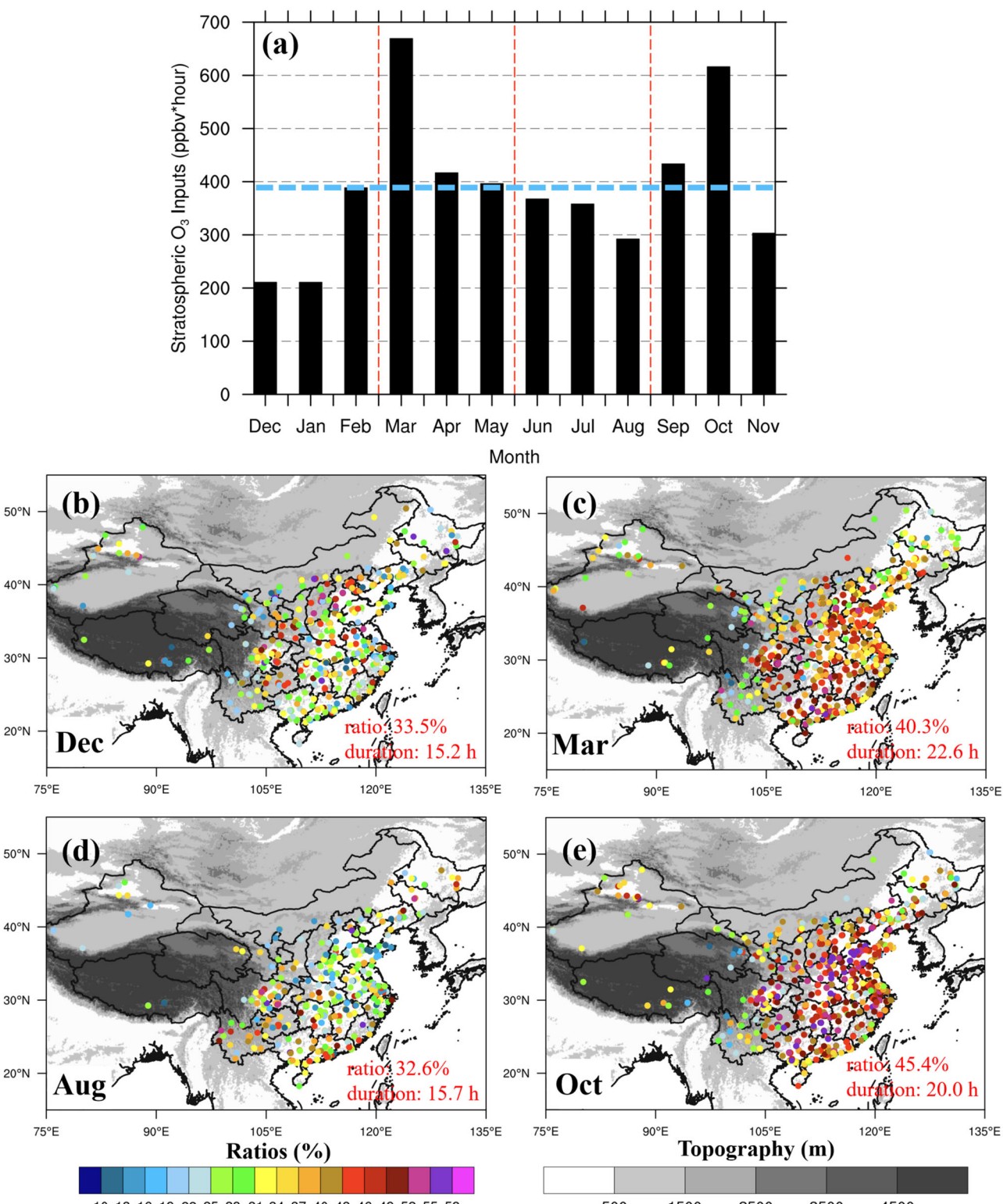

**Fig. 3 | The monthly sum of stratospheric O₃ inputs ($O_3^{strat}$) and the ratios of $O_3^{strat}$ to the sum of surface O₃ concentrations during stratospheric intrusions to the surface (SITS) events ($Ratio_{SITS}$). a** The monthly sum of $O_3^{strat}$ (unit: ppbv*hour; see Eq. (5) in "Methods" section) in China averaged over 2015-2022. The horizontal blue dashed line represents its annual mean. **b**–**e** Spatial distributions of $Ratio_{SITS}$ during the periods of the SITS events (unit: %; see Eq. (4) in "Methods" section) in December, March, August and October. The red numbers in the lower right corner are the mean $Ratio_{SITS}$ and the mean duration of SITS events in the corresponding months on a national scale. Source data are provided as a Source Data file.

enhancement of O₃ and reduction of CO, the two indicators of SITS used in our detection method, are obvious at the moment when SITS events start. Compared with the baseline values representing the non-SITS conditions, surface O₃ is generally enhanced by

20 ppbv in the initial hours when stratospheric O₃ reaches the surface and still retains its stratospheric properties largely, leading to a positive O₃ anomaly of 60−70%. Right after SITS occurrences, the 90th percentile of surface O₃ concentrations is directly enhanced

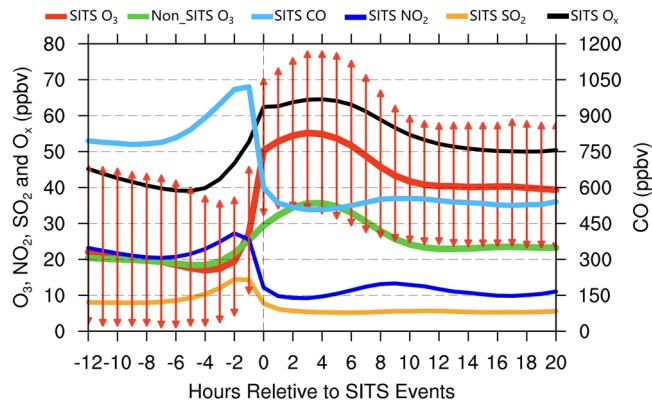

**Fig. 4 | Composite analysis of hourly O₃, CO, NO₂, SO₂ and total oxidants (Oₓ = O₃ + NO₂) concentrations during all detected stratospheric intrusions to the surface (SITS) events in China over 2015-2022.** The zero hour represents the start hour of all SITS events. A period of 33 h is applied starting 12 h before the SITS start hour and ending 20 h after the start hour. O₃ concentrations of each SITS events (SITS O₃, red solid line, unit: ppbv) are aligned into the 33-h period and averaged in each hour. The red arrows measure the 10th and 90th percentile of SITS O₃ concentrations. The same procedure is applied to the hourly O₃ reference baselines (Non_SITS O₃, green solid line), CO (SITS CO, light blue solid line), NO₂ (SITS NO₂, deep blue solid line), SO₂ (SITS SO₂, orange solid line), and Oₓ (SITS Oₓ, black solid line). Source data are provided as a Source Data file.

by up to 40 ppbv above their normal values, which can affect adversely human health and ecosystems.

The recommended O₃ threshold by the World Health Organization (WHO) is 50 ppbv[35], which is measured as a daily maximum 8-h average (MDA8), while the threshold is 70 ppbv in USA[7], and 80 ppbv in China[12]. When MDA8 O₃ is larger than the threshold, O₃ exceedances occur. Figure 5 shows the fraction of SITS events with O₃ exceedances (based on the three O₃ thresholds above) to the total number of SITS events in each month, averaged over all the stations and all the years. Here for each standard, if there is at least one O₃ exceedance during a SITS event, we regard the event as a SITS-induced O₃ exceedance. Referring to the WHO standard (Fig. 5a), over 70% of SITS events are associated with O₃ exceedances from the middle of spring to summer, with an annual mean ratio of 41.1%. Similarly, the inputs of stratospheric O₃ can lead to high O₃ exposure to more harmful-level concentrations, even above the US and Chinese O₃ standards (Fig. 5b, c). In spring, O₃ in the lowermost stratosphere builds up and frequent tropopause folding events take place[36–38]. In addition, the seasonal variation in the surface background O₃ shows an overall maximum in spring and summer in China[39] (Fig. 5d). Under such conditions, air pollution is greatly exacerbated by SITS events from April to September, raising great health concerns especially during the warm months.

For CO variations during SITS periods as shown in Fig. 4, the composite analysis indicates a reduction of 30-35% compared with their values before SITS occurrences. Other gas pollutants that are not used for identifying SITS events here, such as NO₂ and SO₂, also exhibit a prominent decline during the SITS hours. We also examined the responses of atmospheric oxidation capacity (AOC) to SITS using total oxidant (Oₓ=O₃ + NO₂) as an indicator. Despite a reduction of NO₂ when SITS events occur, the injected stratospheric O₃ compensates for the loss of NO₂ and instantly increases the AOC by 20-35%. As a consequence, the enhanced AOC would help stimulate the formation of particulate nitrate and secondary organic aerosols[40]. Stratospheric air injected into the surface can substantially alter the tropospheric air compositions, reaction sensitivity regimes between O₃-NOₓ-VOC and tropospheric oxidative states[34,41,42]. Even though SITS events are transient, the complicated perturbation from the stratosphere induces

changes in the balanced tropospheric air and can amplify the initial effects of SI[18,43–45]. Therefore, SITS should be seriously considered in tropospheric chemistry and air pollution control[46].

## Declining influences of SITS on surface ozone variations in 2015-2022

The quasi-decadal observations enable us to examine the long-term variation of stratospheric influence on surface O₃. Supplementary Fig. 1a shows variations in the monthly stratospheric O₃ inputs to the surface (in the unit of ppbv*hour, see "Methods" section, Eq. (5)) in China over 2015-2022 and their ratio to the overall surface O₃ during SITS periods in the affected areas (see "Methods" section, Eq. (4)). The ratio ranges between 25-50%, and shows substantial interannual variations. Over 2015-2022, the magnitude of direct stratospheric influence on surface O₃ across China appears declining. Figure 6a further shows the time series of deseasonalized monthly accumulated SITS-induced O₃ over 2015-2022. A declining trend significantly at a 95% level is observed at a rate of -6.7 ppbv*hour per year, which is ~1–2% per year. An independent indicator of the stratospheric influence, the occurrence of very-high-O₃ concentrations during SITS (e.g., 80–100 ppbv; Supplementary Fig. 2), also exhibits a decreasing trend, supporting the results of Fig. 6a.

If scaled to longer timescales of a month or year (see Eq. (6) in "Methods" section), the ratio of stratospheric inputs to surface O₃ is much lower (Fig. 6b and Supplementary Fig. 1b). On a annual basis, 1.6-2.2% of surface O₃ is attributable to the stratospheric inputs in the affected areas. The stratospheric O₃ ratios at the surface exhibit a maximum in early spring (2.7%) and autumn (2.2%), but a minimum in the summer (1.3%). Low fractions of stratospheric O₃ inputs to surface O₃ over annual or longer periods have been documented. For example, Cristofanelli et al.[47] estimated that deep SI contributed 2% of surface O₃ on the southern slope of the Himalayan region, and Lin et al.[48] found that the SI contributed 1.3% of surface O₃ in Mt. Hehuan of Taiwan.

Over 2015–2022, the ratio of SITS-induced O₃ to overall surface O₃ concentrations also declined significantly at -0.074% per year (Fig. 6b). The model simulations of Verstraeten et al.[13] suggested increasing stratospheric contributions to the tropospheric O₃ increases in China over 2005–2010. Yet, based on the analysis of surface observations, this study suggests a minor and declining stratospheric influence on the surface O₃ increase in China over 2015–2022, at least for the direct and deep SI events. The causes for the declining trend of stratospheric influences may include the weakened O₃ abundance in lower stratosphere[49,50] (Supplementary Fig. 3a) where STT events take place mostly, and the reduced strong SITS occurrences (Supplementary Fig. 2). Regarding the tropospheric environments where stratospheric O₃ intruded into, the capping stable layer (thermal inversion) tends to descent and intensify since 2010 in China (Supplementary Fig. 3b), hindering the formation of deep PBL for downward transport of stratospheric air to the surface[15,16]. The stratospheric O₃ injected to the troposphere averaged over China, deriving from the Trajectory-mapped Ozonesonde dataset for the Stratosphere and Troposphere (TOST)[51] data, also indicates a declining tendency over 2015-2022 (Supplementary Fig. 4). Supplementary Fig. 5 presents the estimated amounts of stratospheric O₃ in the surface layer from the MERRA-2 GMI global atmospheric chemistry model[52], which also shows a decreasing trend of stratospheric contributions to surface O₃.

## Discussion

Estimates of the stratospheric influences on surface O₃ are necessary for making effective mitigation policies, since these inputs of natural stratospheric O₃ can substantially enhance the risk of O₃ pollution episodes and partially determine the floor value for air quality managements. In a short term, SITS-induced O₃ has non-negligible significance for transient high-O₃ episodes, given its large fractions of surface O₃ budget (30-45%) and high risks of O₃ exposure to harmful-

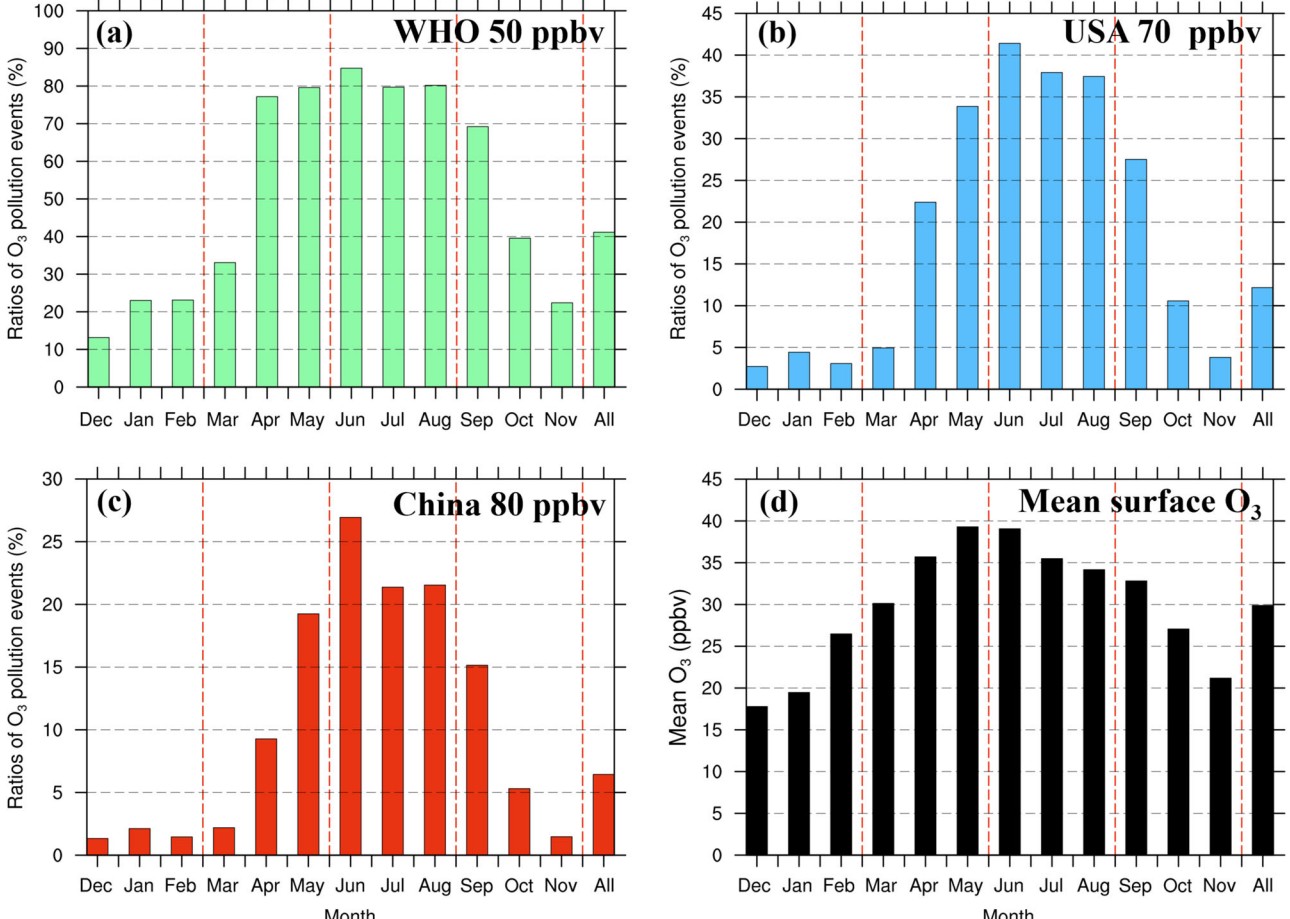

**Fig. 5 | The fraction of stratospheric intrusions to the surface (SITS) events with O₃ exceedance to the total SITS events (unit: %) under different air quality standards and mean surface O₃ concentrations in each month over 2015–2022.** The recommended O₃ standard by the World Health Organization (WHO) is 50 ppbv (**a**) measured as 8-h maximum moving average within a day (MDA8), while it is 70 ppbv (**b**) and 80 ppbv (**c**) in the USA and China, respectively. In each detected SITS event, the MDA8 O₃ is calculated and compared against the standards above to determine the occurrence of O₃ exceedance. **d** Mean surface O₃ concentrations (unit: ppbv) averaged over all surface stations in each month. The monthly variations in the fraction and surface O₃ concentrations are the mean over 2015–2022, while the rightmost column represents the annual mean. Source data are provided as a Source Data file.

level concentrations during SITS periods in affected areas. While the absolute stratospheric influences are highest in March and October, special attention to O₃ pollution control should be paid in spring and summer when extra SITS-induced O₃ inputs, plus the high background O₃, promote possibility to exacerbate O₃ pollution beyond the WHO and national standards. In 10% of the SITS cases, surface O₃ can be elevated by over 40 ppbv, setting alarms for possible severe O₃ pollution in affected areas. On the other hand, SITS could synchronously reduce concentrations of other air pollutants, including CO, NO₂, and SO₂. Such stratospheric perturbation can also substantially enhance the oxidation capacity of tropospheric air. On the annual basis, detectable O₃ with stratospheric origins consists of 1.6–2.2% of surface O₃ in China, implying that O₃ pollution mitigation over long terms in China should mainly focus on surface O₃ variations through photochemical reactions under the influence of meteorology and anthropogenic emissions. In this study, we conservatively estimate the stratospheric influences by only including the dynamically injected stratospheric O₃ associated with deep, direct, and fast SITS events. The influences of aged stratospheric air injected into the troposphere from the stratosphere are spared, since such influences could not be easily identified from surface measurements. A combination of satellite remote sensing technology and deep machine learning methods in future work can help solve these issues. The chemically-induced O₃ production due to stratospheric perturbation may also contribute to

surface O₃ variations in the presence of nonlinear chemical reactions during SITS[53,54]. The above factors can amplify the influences of the stratosphere especially in transient surface O₃ pollution events, and hence enhance the impact of O₃ on human health and crop yield.

## Methods

### Screening SITS based on comprehensive surface observations

Taking advantage of surface gaseous pollutant measurements, e.g., O₃, CO, NO₂, and SO₂, with a high spatial and temporal resolution, here we develop a methodology of detecting SITS events over large areas and for long periods, based on and further refined from our previous studies[9,14]. This method is effective in detecting deep, direct, and fast SI retaining stratospheric properties, such as "O₃-rich and CO-poor"[3,4,15]. In this study, we focus on such SITS events, while aged stratospheric air that has reached the surface but lost its stratospheric properties is not considered. Relying on the characteristics of stratospheric air reaching the surface (richer O₃ and poorer CO relative to tropospheric air), we identify a SITS event based on hourly concurrent O₃ and CO measurements at the surface based on the following points.

1. Distinct upward and downward spikes of O₃ and CO indicators, respectively. The hourly measurements of stratospheric indicators (O₃ and CO) are screened to filter out their distinct spikes, e.g., the sharp increase in O₃ and decrease in CO simultaneously, a unique indication for air with recent stratospheric origins. Hourly

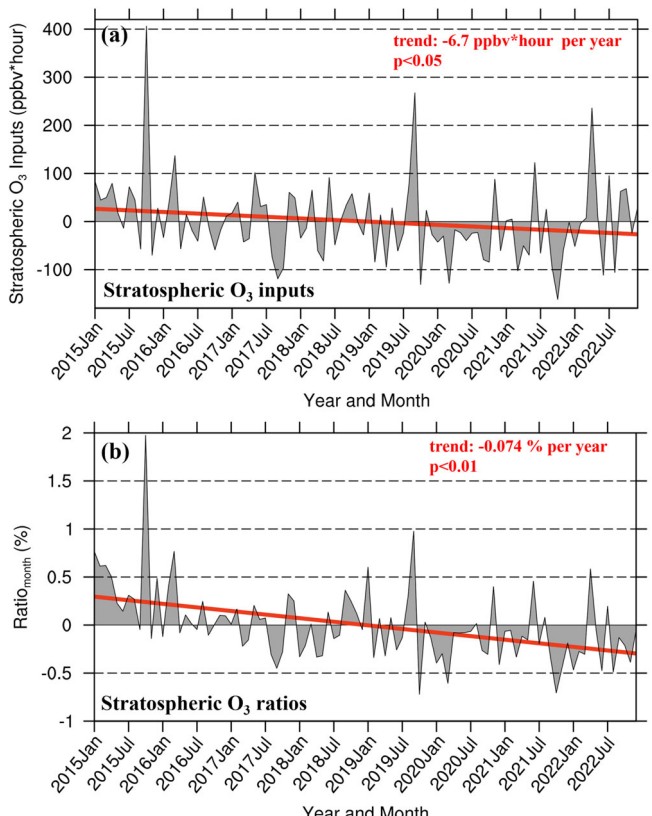

**Fig. 6 | Deseasonalized monthly stratospheric $O_3$ inputs and their ratios to surface $O_3$ concentrations averaged at all study sites in China over 2015-2022.** The gray shaded areas represent the time series of deseasonalized monthly means of (**a**) stratospheric $O_3$ inputs (see Eq. (5) in "Methods" section) and (**b**) their ratios to the surface $O_3$ concentrations (see Eq. (6) in "Methods" section). The red lines represent the linear trends for stratospheric $O_3$ inputs and stratospheric $O_3$ ratios, respectively, and the red numbers in the upper right corner are the trends that are statistically significant above the 95% and 99% confidence level, respectively. Source data are provided as a Source Data file.

variations of $O_3$ and CO concentrations throughout the year are calculated site by site and year by year. The 95th percentile of the $O_3$ rising rate and 5th percentile of the CO decline rate in each year are chosen to identify those sudden and sharp spikes when stratospheric air initially reaches the surface[55]. The synchronous appearance of extreme $O_3$ increase and CO decrease could help isolate the sudden surface $O_3$ change due to SI from that due to $O_3$ transport or photochemical processes. As shown in Supplementary Fig. 6, the two parameters simultaneously determine the start timing of a SITS event (SITS_start).

2. The large departures of $O_3$ and CO from their normal values. The intruded stratospheric air contains higher $O_3$ than that in the troposphere; therefore, surface $O_3$ with additional inputs in SITS events is supposed to increase from its normal values. To minimize the blurring of photochemically produced tropospheric $O_3$, we consider that the $O_3$ concentrations at the SITS_start hour should exceed the seasonal mean value during noontime ($\bar{O}_3^{noon}$, 1st $O_3$ criterion) when photochemical reactions are active. Simultaneously, the CO concentrations during the SITS should decline below their seasonal mean value (CO criterion). These criteria could also help remove the occasions that could be falsely identified when $O_3$ is transported downward from the residual layer, an $O_3$-rich "reservoir" containing photochemically produced $O_3$ in the preceding day[56,57]. Due to the mixing with tropospheric air and chemical sinks of $O_3$, the properties of the intruded stratospheric air subside over the time[58]. When the $O_3$

concentrations fall back to their seasonal mean values ($\bar{O}_3^{season}$, 2nd $O_3$ criterion) or the CO concentrations rebound over the CO criterion, stratospheric air is not distinguishable from the tropospheric air and hence the SITS events end (SITS_end; referred to the case illustrated in Supplementary Fig. 6).

Provided with the start and end timing of SITS events, we estimate the amounts of injected stratospheric $O_3$ reaching the surface by integrating the excess of $O_3$ concentrations above their reference baselines (the seasonal means at the corresponding hour) during the SITS events (see details in the next section). At a given station for a period, such as a month, the number of SITS occurrences, the length of time between SITS_start and SITS_end averaged over all SITS events in the period, and the averaged excess of $O_3$ concentrations above the baselines are regarded as the frequency, duration, and intensity of the SITS at that station for that period.

Similar to the definition of a chemical tropopause[59], we rely on the variations in atmospheric chemical constituents $O_3$ and CO, rather than some dynamic indicators, to define the frequency, duration, and intensity of SITS events. These definitions are referred throughout this manuscript. Both $O_3$ abundance in the lower stratosphere and frequency of deep stratosphere-to-troposphere processes primarily determine the injected amounts of stratospheric $O_3$ into the troposphere[60]. When intruded into the troposphere, stratospheric $O_3$ can be strongly mixed with tropospheric air and be chemically destroyed, responding to the complicated dynamical and chemical processes in the troposphere, especially in the PBL. Therefore, assessing the stratospheric contribution to surface $O_3$ depends on not only the detailed information of SITS (e.g., their frequencies and magnitudes), but also the varying tropospheric environments that control the fate of injected stratospheric $O_3$ (SITS duration).

Although stratospheric air is also characterized with low RH, RH is not selected as an indicator in our detect algorithm. This is because RH is inversely related to temperature. Low RH may also appear when air parcels descend from higher altitudes to the lower troposphere experiencing adiabatic warming. The air parcels can also experience various atmospheric moisture conditions on their way to the surface, so RH of the air parcels is less conservative than $O_3$ and CO[3,8]. In addition, concurrent RH measurements are usually unavailable in air quality monitoring stations in China.

We have developed this SITS methodology with a goal of being objective, robust, and accurate, i.e., reducing the commission and omission errors as much as possible. We have inclined to be conservative and set the detecting criteria rather strictly. For example, we assure that SITS would enhance surface $O_3$ concentrations, i.e., as long as surface $O_3$ concentrations are not above the background value, SITS stops. In this way, the detected SITS events are highly likely to be the cases, while some weak SITS events may be omitted.

### Estimation of contributions of SITS to surface $O_3$

The contributions of SITS to surface $O_3$ are estimated event by event and station by station. The hourly mean surface $O_3$ concentrations ($\bar{O}_3^h$; where h = 1, 2, 3,...24) are calculated by averaging $O_3$ observations at each of the 24 hours in each season based on station-level observations in each year. The $\bar{O}_3^h$ values are taken as reference baselines to measure the departure of $O_3$ concentrations from their baselines during SITS periods. Provided with the start and end timing of SITS events, we integrate the excess of $O_3$ above the reference baselines during SITS periods (unit: ppbv*hour), and take it as the amount of injected stratospheric $O_3$ ($O_3^{strat}$) in each SITS event[14,28,31,48]:

$$O_3^{strat} = \int_{SITS\_start}^{SITS\_end} (O_3^h - \bar{O}_3^h)dt \qquad (1)$$

where $O_3^h$ denotes the in situ hourly $O_3$ observations at hour $h$, and the $\bar{O}_3^h$ represents the baseline $O_3$ concentrations at the same hour. The differences between $O_3^h$ and $\bar{O}_3^h$ are summed over the SITS period with a temporal resolution of 1 hour (i.e., d$t$ = 1 hour).

The sum of $O_3$ concentrations during each SITS event and its corresponding month (unit: ppbv*hour) are calculated following Eqs. (2) and (3), respectively:

$$O_{3SITS}^{sum} = \int_{SITS\_start}^{SITS\_end} O_3^h dt \qquad (2)$$

$$O_{3month}^{sum} = \int_{month\_start}^{month\_end} O_3^h dt \qquad (3)$$

The ratio of stratospheric $O_3$ ($O_3^{strat}$) to the sum of $O_3$ concentrations during each SITS event ($Ratio_{SITS}$) is given by Eq. (4):

$$Ratio_{SITS} = \frac{O_3^{strat}}{O_{3SITS}^{sum}} * 100\% \qquad (4)$$

The sum of stratospheric $O_3$ inputs during all SITS events in a month ($O_{3month}^{strat}$, unit: ppbv*hour) is calculated by Eq. (5), given the number of SITS events in the month being $N$:

$$O_{3month}^{strat} = \sum_{i=1}^{N} O_{3,i}^{strat} \qquad (5)$$

Finally, the ratio of stratospheric $O_3$ to the sum of $O_3$ concentrations in the corresponding month ($Ratio_{month}$) is given by Eq. (6):

$$Ratio_{month} = \frac{O_{3month}^{strat}}{O_{3month}^{sum}} * 100\% \qquad (6)$$

The time series (2015–2022) of SITS-induced $O_3$ and its ratio to overall surface $O_3$ concentrations during SITS periods and the entire month are showed in Supplementary Fig. 1.

## Validations of the SITS detection algorithm

As SITS appears as rare events in local areas, it is important to verify the reliability of our detection algorithm in order to address the impact of deep SI on surface $O_3$. We previously published a detailed analysis of two SITS events that occurred in China, which were selected from the large samples of SITS events[9,14]. The stratospheric origins and transport pathways of the two SITS cases were revealed by means of surface air pollutant observations, vertical profiles of RH and $O_3$, PV evolution, and backward trajectory simulations. The general characteristics of SITS detected with our algorithm, such as their seasonality and contribution to surface $O_3$, are in good agreement with the observed deep SI climatology by Stohl et al.[24], Elbern et al.[28] and Cristofanelli et al.[31]. The detected frequency of SITS is further compared with published observational studies. For example, based on multiple stratospheric tracers including RH, CO, and cosmogenic radionuclide $^7$Be, Lin et al.[48] identified 14 SI days during a 13-month campaign in a high-elevation station (3380 m asl) located in Mt. Hehuan of Taiwan (Fig. 1a). The annual frequencies of SITS in Panzhihua (16 per year) and Chuxiong (14 per year) detected in the present study agree reasonably with those at Mt. Hehuan which is with a similar latitudes and altitudes (Fig. 1a). Using a combination of stratospheric tracers including RH, potential vorticity (PV), $^7$Be and the tropopause height anomaly, Cristofanelli et al.[31] reported that there were 33 days (average 5.5 days per year) of deep and direct SI which were characterized by distinct stratospheric

properties at Mt. Cimone (2165 m asl) in Italy over 1998-2003. We apply the SITS detection algorithm to the $O_3$ and CO measurements collected in Mt. Cimone during 2013–2016, and find a total of 26 direct SI events (average 6.5 days per year). The results from our detection algorithm are in line with these SI studies shown above and indicate the feasibility of using sudden and sharp spikes of $O_3$ and CO to identify SI reaching the surface.

The origins of SITS events are investigated with backward trajectory simulations of 10 days over selected cities (Supplementary Fig. 7; see details of the backward trajectory simulations in the following section). We select Panzhihua as an example where the highest SITS frequency is detected. The cities Beijing and Fuzhou are also selected to examine SITS occurred in northern and southern China, respectively. The trajectory analysis indicates that the majority of air parcels at the surface during the detected SITS events originated in the upper troposphere and lower stratosphere (UTLS; above 400 hPa), i.e., 96% in Panzhihua, and 100% in Fuzhou and Beijing. As another piece of evidence, the ensemble RH profiles during the 33 SITS events over Beijing show substantial dryness characterized by RH < 30% in the PBL and near the ground level[15], suggesting the dry stratospheric air has descended into the surface. In addition to these selected cities, we further analyze the ensemble of backward trajectories associated with the detected 27,616 SITS events (Supplementary Fig. 8). We evenly divide every trajectory from the beginning to the end into three travel stages. The height, PV, and $O_3$ concentrations in each stage are extracted from MERRA-2 reanalysis data. The air parcels of SITS initially reside in 300-200 hPa, where PV values exceed 2 PVU (an iso-surface representing the dynamical tropopause) and $O_3$ concentrations are larger than 250 ppbv[61,62], showing prominent stratospheric origins.

## Backward trajectory simulations and MERRA-2 reanalysis data

Backward trajectories are simulated to check the origins of airmass of detected SITS events (Supplementary Fig. 7 and Supplementary Fig. 8) using the Hybrid Single-Particle Lagrangian Integrated Trajectory (HYSPLIT) model. HYSPLIT is developed by National Oceanic Atmospheric Administration's (NOAA)[63] (https://www.arl.noaa.gov/hysplit). The 10-day backward trajectories are driven by the meteorological data from the Global Forecast System (GFS) with a resolution of 0.25°. The PV values and $O_3$ concentrations along the trajectory are extracted from the Modern-Era Retrospective analysis for Research and Applications version 2 (MERRA-2) reanalysis data. The MERRA-2[64] reanalysis data have a spatial resolution of 0.5° latitude × 0.625° longitude with 72 model levels (https://gmao.gsfc.nasa.gov/reanalysis/MERRA-2; DOI: 10.5067/WWQSXQ8IVFW8).

## Surface observational data and stratospheric $O_3$ tracer data

The present study is mainly based on analysis of hourly ground-based measurements of $O_3$, CO, $SO_2$, and $NO_2$ concentrations at more than 1,600 stations in Chinese cities. For $O_3$ and CO, they are measured with a CO analyzer (Thermo Fisher Model 48i) and an O3 analyzer (Thermo Fisher Model 49i). The detection limit (precision) for Model 48i and Model 49i are 0.04 ppmv (±0.1 ppmv) and 0.5 ppbv (±1 ppbv), respectively. The data are from the public website of the Chinese Ministry of Ecology and Environment (MEE) (https://english.mee.gov.cn/).

To explore the variations in stratospheric $O_3$ during the study period (Supplementary Fig. 3a), the stratospheric $O_3$ profile observations are acquired from the Stratospheric Water and OzOne Satellite Homogenized (SWOOSH) dataset[65] (https://csl.noaa.gov/groups/csl8/swoosh). SWOOSH provides a merged record of stratospheric $O_3$ on the basis of a number of limb sounding and solar occultation satellites from 1984 to the present.

To investigate the thermal inversion variations over China during 2015-2022 (Supplementary Fig. 3b), radiosonde observations in China are processed, which are available from https://www.ncei.noaa.gov/products/weather-balloon/ integrated-global-radiosonde-archive.

The Trajectory-mapped Ozonesonde dataset for the Stratosphere and Troposphere (TOST)[49,66] is a 3-dimensional O₃ dataset derived from ozonesondes at over 100 stations using a trajectory-based mapping methodology with the HYSPLIT model. The thermal tropopause height is determined for each O₃ profile, and the stratospheric O₃ distribution is mapped for the O₃ with stratospheric origination along the trajectory paths. All O₃ values along the trajectory paths are binned into grids of 5° × 5° × 1 km (latitude, longitude, and altitude) from sea level to 26 km in each month. TOST has been validated against independent ozonesondes and widely used in global O₃ climatology studies[67]. In the present study, we further extend the TOST data by conducting 10-day forward trajectories simulations over 2015-2021 (Supplementary Fig. 4).

Simulations from the MERRA-2 GMI global chemical transport model are analyzed for long-term variations in stratospheric O₃ inputs to the surface-layer during the period (Supplementary Fig. 5). The MERRA-2 GMI (Global Modeling Initiative's) model with stratosphere-troposphere chemical mechanisms is driven by MERRA-2 meteorology including winds, temperature and pressure[52] (https://acd-ext.gsfc.nasa.gov/Projects/GEOSCCM/MERRA2GMI/). This model is run at a MERRA-2 native horizontal resolution of ~50 km with 72 vertical levels. The model applies a stratospheric O₃ tracer to diagnose the stratospheric O₃ influence on the troposphere.

## Data availability
The datasets used in this study are freely available and are available from the corresponding authors on request. All data supporting the findings of this study are available within the paper and are provided as a Source Data file. Source data are provided with this paper.

## Code availability
All of the figures are created by the authors using the NCAR Command Language Version 6.5.0 (available at http://www.ncl.ucar.edu/). The codes used in this study are available from the corresponding authors on request.

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

## Acknowledgements

This work was supported by the National Natural Science Foundation of China (Grant No. 42105079 to Z. C.). The computing resources were provided by Fujian Normal University High Performance Computation Center (FNU-HPCC). We appreciate the Chinese Ministry of Ecology and Environment for setting up the nationwide observation network and publishing air quality data. We also thank NASA GMAO for the MERRA-2 reanalysis data, GMI simulations, and NOAA for the SWOOSH data and HYSPLIT model. We are grateful to colleagues and friends, especially Dr. Wenyu Li at University of Toronto, Prof. Jianchun Bian and Dr. Dan Li at Chinese Academy of Sciences, Prof. Lei Wang at Fudan University, for helpful discussions.

## Author contributions

Z.C. and J.L. conceived the overall concept and wrote the manuscript. Z.C. developed the SITS detection code and analyzed the data. X.Q. ran the field campaign for SITS observations, provided the field observations, and helped with the data interpretation. X. C., M. Y., L. S. and Z. Z. performed the related analyses, contributed to discussions, and edited the manuscript.

## Competing interests

The authors declare no competing interests.
