## [Peer Review File · Nature Communications]

Stratospheric influence on surface ozone pollution in ChinaREVIEWER COMMENTS

Reviewer #1 (Remarks to the Author):

It is very interesting and valuable for carrying SITS on China ozone pollution control, specially analyzing from frequency, duration, and intensity within a relative long time. The methods, results and discussion are written well, some minor comments need pay attention.

1. Why select 2015-2022 period? a longer time will be better, for instance 2010 or 2012 to 2022.
2. Satellite remote sensing technology has advantages and can play important role for detecting SITS events, for example, using Sentinel-5P-Tropomi payload, many literatures have documented it, it is hence proposed in further work, adding the description of using satellite means to strengthen monitor and even warn ozone exceedance SITS in combination with atmospheric models, ground observation data and deep machine learning methods.

Reviewer #2 (Remarks to the Author):

Chen et al. quantified the impacts of stratospheric intrusions (SI) on surface ozone in China over 2015-2022 by analyzing three metrics of frequency, duration and intensity. They selected gaseous pollutants of O₃ and CO as indicators to develop a method for detecting stratospheric intrusions to the surface (SITS) events. Results show that SITS events preferentially occur in high-elevation regions, while those occurring in plain regions are more intense, which exacerbate O₃ pollution. SITS-related O₃ magnitudes peak in spring and autumn, and SITS-related O₃ shows a declining trend during 2015-2022 in China. This topic is interesting as the impacts of stratospheric influences on surface O₃ is non-negligible. However, after reading the manuscript, there are some questions which should be explained in details.

1. In the Introduction section (Lines 33-56), authors list many reasons to emphasize the difficult to assess the impact of SITS on surface O₃, such as the impacts of chemical processes in PBL and the physical processes including large-scale tropopause folding and small-scale PBL mixing. According to the developed algorithm in Method section, can the methodology using the concentrations of O₃ and CO alone solves these difficulties? Please explain.
2. More information about the importance for researching the impacts of SITS on O₃ should be added in the manuscript, as the authors repeatedly emphasize that SITS events are transient and extremely rare.
3. It may be better to merge the content of the fifth paragraph (Lines 57-62) into the second paragraph (Lines 28-32).
4. Line 64: how is the change in relative humidity?
5. Lines 95-96: As the authors find that unlike the spatial patterns of frequency and duration, the surface O₃ enhancements are larger in plain than in high-elevation areas. More reasons should be added to explain why larger impacts of SITS on O₃ are calculated over plain regions where less SITS events are identified.
6. Lines 118-121: As the authors declare that the SITS events with O₃ enhancements exceeding the

surface O₃ baseline by less than 15 ppbv (weak SITS) exhibit a maximum in cold months, while those with an exceedance over 40 ppbv (strong SITS) appear frequently in warm months, especially in summer. More explanations should be added for this phenomenon. As shown in Figure 2(a-c), comparing with the SITS events in spring, less events are detected in summer and why the impacts are larger?

7. In Figure 4, the concentration of O₃ is hourly value. So the authors should apply the hourly O₃ threshold for analysis, rather than the threshold of MDA8 O₃.

8. Authors should declare how they define "SITS days". All we find in the manuscript is "SITS events".

9. Lines 265-167: "The 95th percentile of O₃ rising rate and 5th percentile of CO decline rate in each year are chosen to identify those sudden and sharp spikes when stratospheric air initially reaches the surface". Any reason for choosing the thresholds of "95th" and "5th", please explain.

10. Lines 273-275: "To minimize the blurring of photo-chemically produced tropospheric O₃, we consider that the O₃ concentrations at the SITS_start hour should exceed the seasonal mean value during noontime when photochemical reactions are active". Any reason for this treatment, please explain. The process of advection can also make the O₃ concentration exceed or below the seasonal mean. Meanwhile, why choose the seasonal mean value as the threshold?

11. Line 304: Why the absolute concentration must satisfy the predefined criteria?

12. Lines 344-347: The period Cristofanelli et al. analyzed is 1998-2003, while the period the authors analyzed is 2013-2016. It may be hard to compare the results directly to confirm the algorithm in line with the previous studies.

13. In Extended Data Fig. 2, why the authors choose the regions of Panzhihua, Fuzhou, and Beijing as example, please explain.

14. As the authors explained that stratospheric air is characterized by high O₃, low CO and low RH, why only apply O₃ and CO to construct the algorithm?

Reviewer #3 (Remarks to the Author):

Review of "Large stratospheric influence on surface ozone pollution in China" by Chen et al., submitted to Nature Communications.

Summary:

In this manuscript, the authors evaluate surface observations from ground-monitoring sites around China for the period from 2015-2022 in order to identify periods of time when the surface concentrations can be characterized as likely directly influenced by recent transport of stratospheric air (ozone-rich, carbon monoxide-poor, dry air relative to tropospheric conditions). The authors provide detailed description of how they can use the known conditions of stratospheric air to determine when the observations are anomalously different from seasonal background and anthropogenic polluted air. Using this technique, they identified over 27K stratospheric-intrusion-to-surface (which they coin as "SITS") events, which on average enhance surface ozone by 20 ppbv, and can be linked to enhanced ozone pollution conditions during spring and summer.

While I do not think I have seen such a method used on such a grand scale, or for China, the abstract

may oversell the frequency of such events, as this is for a dense network of over 1600 surface stations so the highest frequency rate is around 15 SITS events per year in cities close to the Tibetan Plateau, which is not a surprise given the high elevation i.e., closer to the stratosphere (much like we see for stratospheric intrusion influences for communities in the Rocky Mountains, USA). Looking at Figure 1a, most of China experiences less than 4 SITS events per year (green to blue/grey colors). And yet, in Figure 1c, the ozone enhancement is greatest further east.

I was surprised to see the dominance of the impact in the fall in early figures like Figure 3, as I was expecting to see patterns like the seasonal variation in Figure 5, showing greatest impact of high ozone to surface ozone concentrations in spring and early summer (as discussed in lines 159-162). My understanding is that this because in late winter to early spring there is a build-up of ozone in the lowermost stratosphere so when we have these strong low-pressure systems with tropopause folding events occurring behind the cold front, there is more ozone available with which to be drawn down within the fold (see Danielsen & Mohnen, 1977, Holton et al., 1995, and especially Monks, 2000). I am less familiar with the argument the authors provided (from Trickl et al. 2020) about air parcels coming from higher origins in the stratosphere in summer than other seasons which leads to the maximum in stratospheric ozone intensity.

Additional references to consider are two papers which have studied stratospheric intrusions which impact surface ozone in the Hong Kong region:

Zhao et al. <https://doi.org/10.1016/j.atmosres.2020.105158>

Zhao, K., Huang, J., Wu, Y., Yuan, Z., Wang, Y., Li, Y., et al. (2021). Impact of stratospheric intrusions on ozone enhancement in the lower troposphere and implication to air quality in Hong Kong and other South China regions. *Journal of Geophysical Research: Atmospheres*, 126, e2020JD033955. <https://doi.org/10.1029/2020JD033955>

And one that looks at dominant circulation patterns over parts of China
Yin, Z., Cao, B., and Wang, H.: Dominant patterns of summer ozone pollution in eastern China and associated atmospheric circulations, *Atmos. Chem. Phys.*, 19, 13933–13943, <https://doi.org/10.5194/acp-19-13933-2019>, 2019.

I would have liked to have seen the analysis include more links to the upper-air flow. The authors reference previous works (Line 347) but this requires the reader to go beyond the current manuscript to be convinced that the anomalous surface conditions are indeed connected to the stratosphere. Are there any other meteorological conditions with specific emission environments which could have led to this type of ozone-rich/CO-poor measurements? If the authors can prove this to me, I'd be more convinced that surface-measurements alone are sufficient. Otherwise, a case study showing how the method described on lines 265-270 works with the upper air dynamical features would go a long way.

The manuscript is well written. I have the following minor and technical suggestions.

Line 11: change "to which" to "to what extent"

Line 55: The reference that this has become a highly debatable issue is from twenty years ago, and only

one is provided. That does not seem very debatable if just one reference without any more recent debate.

Line 85: change “locate” to “are located”

Line 88: It would be helpful if elevation of these cities is quoted in the main text.

Line 94: More than reference #6 would be applicable here.

Line 116: It is at this stage I am wondering how these ozone enhancements have any connection to health impacts to motivate the study and the authors do not explain this till much later (line 153-162). I would encourage the authors to consider restructuring their message to keep the readers engaged in the motivation for the study.

Line 142: Was a statistical test performed to determine that the enhancement was “significant”?

Line 150: change “averagely” to “generally”

Line 157: change “sever” to “severe”

Line 158: Probably better to say over 70 % in warm months if considering March through August.

Line 171: Provide the Figure reference to the end of the sentence.

Line 179: write out stratospheric intrusion, in place of “SI”.

Line 204: Should Figure 6b also be referenced along with Supp. Figure 1b.

Line 205-207: Where are the numbers coming from quoted here? Is it off a figure? If it is coming from a supplemental figure, then the figure should be considered for the main text as substantive results should not be given from supplemental material.

Line 215-216: I do not follow the authors’ connection here to the monthly accumulated ozone pollution hours during SITS in the supplemental figure 2. Likely the wrong figure is referenced here.

Line 317: Add a space before “and”

Line 401-402: MERRA-2 reanalysis data has DOIs per each file collection. These should be included in manuscripts.

References:

Danielsen, E. F., & Mohnen, V. A. (1977). Project dustorm report: Ozone transport, in situ measurements, and meteorological analyses of tropopause folding. *Journal of Geophysical Research*, 82(37), 5867–5877. <http://dx.doi.org/10.1029/JC082i037p05867>

Holton, J. R., Haynes, P. H., McIntyre, M. E., Douglass, A. R., Rood, R. B., & Pfister, L. (1995). Stratosphere-troposphere exchange. *Reviews of Geophysics*, 33(4), 403–439. <https://doi.org/10.1029/95RG02097>

Monks, P. S. (2000). A review of the observations and origins of the spring ozone maximum. *Atmospheric Environment*, 34(21), 3545–3561.

Review submitted by K. Emma Knowland

REVIEWER COMMENTS

Reviewer #1 (Remarks to the Author):

It is very interesting and valuable for caring SITS on China ozone pollution control, specially analyzing from frequency, duration, and intensity within a relative longtime. The methods, results and discussion are written well, some minor comments need pay attention.

1. Why select 2015-2022 period? a longer time will be better, for instance 2010 or 2012 to 2022.

Response: Thank you for this question. Certainly, a longer period would better depict the temporal variation and spatial distribution of SITS events in China. However, it is a pity that surface ozone measurements in China were widely implements only since 2015. Therefore, we only select the 2015-2022 period to screen out SITS events observed in ground-based stations where 8-year continuous ozone and CO measurements are available.

2. Satellite remote sensing technology has advantages and can play important role for detecting SITS events, for example, using Sentinel-5P-Tropomi payload, many literatures have documented it, it is hence proposed in further work, adding the description of using satellite means to strengthen monitor and even warn ozone exceedance SITS in combination with atmospheric models, ground observation data and deep machine learning methods.

Response: Agreed. Satellite data provide an extra monitoring tool to study the occurrence of SITS events over broad areas. Essentially, it offers more information of the vertical evolution of such events in addition to the surface network observations. A combination of surface ozone behavior relying on ground-based dense monitoring network and vertical evolution based on satellite data would greatly deepen our understanding of where and when the stratospheric intrusions occur. Provided with these knowledge and multi-source data, it is feasible to diagnose and predict the stratospheric intrusions with the use of deep machine learning methods. Thanks for these insightful suggestions

that sheds light on our future work! We have added these in the revision (Line 258-259).

Reviewer #2 (Remarks to the Author):

Chen et al. quantified the impacts of stratospheric intrusions (SI) on surface ozone in China over 2015-2022 by analyzing three metrics of frequency, duration and intensity. They selected gaseous pollutants of O₃ and CO as indicators to develop a method for detecting stratospheric intrusions to the surface (SITS) events. Results show that SITS events preferentially occur in high-elevation regions, while those occurring in plain regions are more intense, which exacerbate O₃ pollution. SITS-related O₃ magnitudes peak in spring and autumn, and SITS-related O₃ shows a declining trend during 2015-2022 in China. This topic is interesting as the impacts of stratospheric influences on surface O₃ is non-negligible. However, after reading the manuscript, there are some questions which should be explained in details.

1. In the Introduction section (Lines 33-56), authors list many reasons to emphasize the difficult to assess the impact of SITS on surface O₃, such as the impacts of chemical processes in PBL and the physical processes including large-scale tropopause folding and small-scale PBL mixing. According to the developed algorithm in Method section, can the methodology using the concentrations of O₃ and CO alone solves these difficulties? Please explain.

Response: Thank you for raising this good question. For those extremely deep stratospheric intrusions reaching the ground level (i.e., SITS in this paper), they have to overcome two barriers, the tropopause and the PBL, and hence encompass multi-scale dynamical processes. Such complicated processes are quite challenging for models to capture. On the other hand, the stratospheric air is typical of high O₃ and low anthropogenic pollutant (such as CO) compared with the tropospheric and surface air. If stratospheric air parcels do reach the surface and still contain some stratospheric features that are distinguishable from tropospheric air, they would result in spikes in the measured surface O₃

and CO, yielding the stratospheric impact at ground level. In other words, the sharp and vigorous changes both in surface O₃ and CO concentrations can be utilized as stratospheric tracers. Therefore, provided with dense surface-based observations of O₃ and CO with a high temporal resolution (1 hour here), we can screen out the rapid and vigorous variations of O₃ and CO concentrations to detect concurrent spikes associated with SITS by means of strict thresholds. This method bypasses the need for detailed knowledge about the multi-scale dynamical processes and complicated chemical sinks for the descending stratospheric air. We have added this point in the revision (Line 65-72 and Line 265-314). The altitudes and meteorological conditions associated with the descending routes of the SITS events are also statistically analyzed in Extended Data Fig. 4 to validate our method.

2. More information about the importance for researching the impacts of SITS on O₃ should be added in the manuscript, as the authors repeatedly emphasize that SITS events are transient and extremely rare.

Response: Yes, we agree. Despite its rareness, SITS is important because it can directly affect the surface O₃ concentrations causing severe O₃ pollution and damaging human health, e.g., Figure 5. Thank you for your suggestion. We have modified the expression and emphasis the importance of SITS in terms of pollution and human health (Line 36-45).

3. It may be better to merge the content of the fifth paragraph (Lines 57-62) into the second paragraph (Lines 28-32).

Response: Thanks for this good suggestion. We have modified it in the revised manuscript.

4. Line 64: how is the change in relative humidity?

Response: As shown in Extended Data Fig. 1 and many case studies of stratospheric intrusions, relative humidity (RH) drops when stratospheric air

reaches the surface. Yet, it is still debatable to use RH as a stratospheric tracer, since RH is inversely related with temperature. Low RH may also appear when air parcels descend from higher altitude to lower troposphere experiencing adiabatic warming (Stohl et al., 2000). Also, the air parcels can experience various atmospheric moisture conditions on their way to the surface so that RH of the air parcels is less conservative than O₃ and CO. Thus, applying RH as a stratospheric tracers may increase the uncertainty of the detected SITS. We have discussed this in Line 310-314 in this revision.

Reference: Stohl, A., Spichtinger-Rakowsky, N., Bonasoni, P., Feldmann, H., Memmesheimer, M., Scheel, H., Trickl, T., Hübener, S., Ringer, W., and Mandl, M.: The influence of stratospheric intrusions on alpine ozone concentrations, *Atmos. Environ.*, 34, 1323–1354, [https://doi.org/10.1016/S1352-2310\(99\)00320-9](https://doi.org/10.1016/S1352-2310(99)00320-9), 2000.

5. Lines 95-96: As the authors find that unlike the spatial patterns of frequency and duration, the surface O₃ enhancements are larger in plain than in high-elevation areas. More reasons should be added to explain why larger impacts of SITS on O₃ are calculated over plain regions where less SITS events are identified.

Response: Thank you for your suggestion. The thresholds for SITS for calculated station by station. Results show that more SITS occurred in high-elevation regions. While SITS events were less frequent detected in plain regions compared with those in high-elevation regions, they were intense inducing larger O₃ jump. The reason for this can be subject to further study. Here, we suspect that for those SITS events descending to plain regions, they have to travel a deeper vertical extension experiencing more chance to be diluted by tropospheric air. Hence, only those strong SITS are more likely to reach the surface over plain regions. We have added this explanation in the revision (Line 100-106). Trickl et al. (2020) suggested that the intensity of O₃ enhancements during the SITS partially depend on how high the intrusions start

in the stratosphere. Possibly, SITS events over plain regions in eastern China may initiate at higher altitudes within the stratosphere.

Reference: Trickl, T., Vogelmann, H., Ries, L., and Sprenger, M.: Very high stratospheric influence observed in the free troposphere over the northern Alps – just a local phenomenon?, *Atmos. Chem. Phys.*, 20, 243–266, <https://doi.org/10.5194/acp-20-243-2020>, 2020

6. Lines 118-121: As the authors declare that the SITS events with O₃ enhancements exceeding the surface O₃ baseline by less than 15 ppbv (weak SITS) exhibit a maximum in cold months, while those with an exceedance over 40 ppbv (strong SITS) appear frequently in warm months, especially in summer. More explanations should be added for this phenomenon. As shown in Figure 2(a-c), comparing with the SITS events in spring, less events are detected in summer and why the impacts are larger?

Response: The ratios of SITS with different intensities of ozone enhancement (weak, moderate and strong) were calculated in each month as shown in Fig 2d. Though the frequency of SITS events is less in summer than in other seasons, a larger portion of them are strong SITS that can enhance surface ozone by over 40 ppbv. As known, stratospheric impact is noticeable only if the stratospheric air at ground level is distinguishable from tropospheric air. The tropopause height is higher in summer than in other seasons in the Northern Hemisphere extratropics (Birner et al., 2006). Therefore, the SITS events in summer originate at higher altitudes and are closer to the altitudes where stratospheric O₃ reaches the maximum around 20-25 km. As a result, strong SITS events carrying higher stratospheric O₃ are more prominent in summer. We have added more explanation in the revision (Line 129-131).

Reference: Birner, T.: Fine-scale structure of the extratropical tropopause region, *J. Geophys. Res.*, 111, D04104, <https://doi.org/10.1029/2005jd006301>, 2006.

7. In Figure 4, the concentration of O₃ is hourly value. So the authors should

apply the hourly O₃ threshold for analysis, rather than the threshold of MDA8 O₃.

Response: Thank you for pointing this out. Indeed, Figure 4 is intended to show the vigorous SITS-induced jump in surface O₃ concentrations revealed by the composite analysis using hourly O₃ concentration values. In Figure 5, we further examine the health effects brought by SITS. In literature daily MDA8 thresholds are mostly used and can provide an assessment over entire SITS period, not individual hours. We have explained the use of daily MDA8 thresholds in this revision in Line 169-179.

8. Authors should declare how they define "SITS days". All we find in the manuscript is "SITS events".

Response: Sorry for this. In the last version, SITS days refers to days with SITS events. In this revision, we have removed this term and used SITS events throughout to avoid confusion.

9. Lines 265-167: "The 95th percentile of O₃ rising rate and 5th percentile of CO decline rate in each year are chosen to identify those sudden and sharp spikes when stratospheric air initially reaches the surface". Any reason for choosing the thresholds of "95th" and "5th", please explain.

Response: Theoretically, if the stratospheric air descends to ground level, the stratospheric properties of the air (high in O₃ and low in CO) would induce prominent changes in the hourly O₃ and CO measurements compared with tropospheric air. Very extreme thresholds for hourly O₃ and CO variations would unambiguously screen out SITS events but miss those with moderate-to-weak magnitudes. Based on our previous study with analysis of high frequency O₃ and CO measurements, radar and satellite vertical measurement of RH and O₃, and WRF model simulations, we found the thresholds of "95th" for O₃ increase and "5th" for CO reduction at ground level are strict enough to screen out SITS events, which also showed reasonable frequency of SITS reported in other SI climatology studies (see Methods). We have added more discussion on these

criteria in this revision (Line 277-278 and Line 310-314).

10. Lines 273-275: "To minimize the blurring of photo-chemically produced tropospheric O₃, we consider that the O₃ concentrations at the SITS_start hour should exceed the seasonal mean value during noontime when photochemical reactions are active". Any reason for this treatment, please explain. The process of advection can also make the O₃ concentration exceed or below the seasonal mean. Meanwhile, why choose the seasonal mean value as the threshold?

Response: The photochemically produced O₃ associated with PBL processes can also induce large surface O₃ variations. During the daytime, high O₃ concentrations through photochemical reactions are distributed in the mixing layer. During the nighttime, a shallow stable boundary layer forms near the surface, and the remnants of the daytime mixing layer form the residual layer containing O₃-rich air mass. The air in the residual layer can be transported downward during the establishment of daytime mixing layer in the next and hence leads to large O₃ variations. In addition to such vertical circulation, the horizontal advection of O₃ can also lead to higher or lower O₃ concentrations relative to their seasonal mean. Here, we assume that the descending stratospheric air contains higher levels of O₃ compared with those from active photochemical reactions in noontime. Therefore, in the initial hour when stratospheric air reaches the surface, the hourly O₃ concentrations at that moment should exceed their seasonal mean values during noontime. In addition to high O₃ concentration values, a sharp reduction in CO should also be satisfied, which excludes the possibility of the advection of anthropogenic polluted air containing both high O₃ and CO. Since surface O₃ shows prominent seasonality, the thresholds are calculated season by season. These seasonal thresholds (baseline values for O₃ and CO) measure the time when stratospheric features fade and are no longer distinguishable from the normal tropospheric air. In developing the detecting SITS methodology, we incline to be conservative and set the detecting criteria strict. In this way, the detected SITS events are highly

likely to be the cases, while some weak SITS events may be omitted. We have added this discussion in this revision (Line 310-314).

11. Line 304: Why the absolute concentration must satisfy the predefined criteria?

Response: In the SITS detection algorithm, both relative and absolute changes in O₃ and CO must be satisfied. The sharp spikes of O₃ and CO are determined by their "95th" and "5th" variations. The requirement of absolute concentrations to exceed their seasonal mean values guarantees that the stratospheric intrusion results in higher O₃ and lower CO than their baselines so that the air parcels really come from the stratosphere, and they are distinguishable from tropospheric air. We explain that we have developed this SITS methodology with a goal of being objective, robust, and accurate, i.e., reducing the commission and omission errors as much as possible. We incline to be conservative and set the detecting criteria strict. In this way, the detected SITS events are highly likely to be the cases, while some weak SITS events may be omitted.

12. Lines 344-347: The period Cristofanelli et al. analyzed is 1998-2003, while the period the authors analyzed is 2013-2016. It may be hard to compare the results directly to confirm the algorithm in line with the previous studies.

Response: Yes, we agree. The number of stratospheric intrusions depends on how they are detected using different datasets and methods. The results of Cristofanelli et al. were based on a combination of multiple stratospheric tracers, which can increase the reliability of detection. Therefore, the annual frequency over a long period in their research can be used as a constrain for comparisons. Limited by data availability, we can only calculate frequency of SITS over 2013-2016 in Mt. Cimone based on our method. The annual mean frequency of SITS detected in the same site was in good agreement with Cristofanelli et al., suggesting that our detection method produced reasonable outputs. This

provided a semi-quantitative assessment. In addition to these comparisons, we also analyzed the trajectories of SITS to further validate our method (Please refer to the Methods section). In the revision, we added more detailed validation results (Line 387-391). With the new information, we hope the reviewers and readers can be convinced by the robustness of our detect algorithm.

13. In Extended Data Fig. 2, why the authors choose the regions of Panzhihua, Fuzhou, and Beijing as example, please explain.

Response: Thank you for pointing this out. As there are over 1500 stations applied in this study, it is not feasible to provide trajectory results of all stations. Hence, we show the origin height of air parcels over Panzhihua where the highest SITS frequency is detected. The cities Beijing and Fuzhou were selected as examples of SITS occurred in northern and southern China, respectively. We have added more explanation about these choices in the revision.

14. As the authors explained that stratospheric air is characterized by high O₃, low CO and low RH, why only apply O₃ and CO to construct the algorithm?

Response: Stratospheric air is generally devoid of most anthropogenic pollutants found in the troposphere such as CO, but rich in O₃. Therefore, we screen out simultaneous spikes in O₃ and CO to detect SITS over stations where both O₃ and CO were continuously measured. We exclude RH in the detection algorithm because the values of RH depend on temperature and they are much variable as explained in Comments 4. Also, the comprehensive observational dataset provided by the Chinese Ministry of Ecology and Environment does not contain concurrent RH information. For the reasons above, we do not apply RH as a stratospheric tracer to reduce the uncertainty of detected SITS. We have discussed this in revision (Line 310-314).

Reviewer #3 (Remarks to the Author)

Review of “Large stratospheric influence on surface ozone pollution in China” by Chen et al., submitted to Nature Communications.

Summary:

In this manuscript, the authors evaluate surface observations from ground-monitoring sites around China for the period from 2015-2022 in order to identify periods of time when the surface concentrations can be characterized as likely directly influenced by recent transport of stratospheric air (ozone-rich, carbon monoxide-poor, dry air relative to tropospheric conditions). The authors provide detailed description of how they can use the known conditions of stratospheric air to determine when the observations are anomalously different from seasonal background and anthropogenic polluted air. Using this technique, they identified over 27K stratospheric-intrusion-to-surface (which they coin as “SITS”) events, which on average enhance surface ozone by 20 ppbv, and can be linked to enhanced ozone pollution conditions during spring and summer.

While I do not think I have seen such a method used on such a grand scale, or for China, the abstract may oversell the frequency of such events, as this is for a dense network of over 1600 surface stations so the highest frequency rate is around 15 SITS events per year in cities close to the Tibetan Plateau, which is not a surprise given the high elevation i.e., closer to the stratosphere (much like we see for stratospheric intrusion influences for communities in the Rocky Mountains, USA). Looking at Figure 1a, most of China experiences less than 4 SITS events per year (green to blue/grey colors). And yet, in Figure 1c, the ozone enhancement is greatest further east.

Response: Thank you for your review of our manuscript and for your detailed understanding of this work.

I was surprised to see the dominance of the impact in the fall in early figures like Figure 3, as I was expecting to see patterns like the seasonal variation in Figure 5, showing greatest impact of high ozone to surface ozone

concentrations in spring and early summer (as discussed in lines 159-162). My understanding is that this because in late winter to early spring there is a build-up of ozone in the lowermost stratosphere so when we have these strong low-pressure systems with tropopause folding events occurring behind the cold front, there is more ozone available with which to be drawn down within the fold (see Danielsen & Mohnen, 1977, Holton et al., 1995, and especially Monks, 2000). I am less familiar with the argument the authors provided (from Trickl et al. 2020) about air parcels coming from higher origins in the stratosphere in summer than other seasons which leads to the maximum in stratospheric ozone intensity.

Response: We have discussed the impact of SITS in terms of its absolute magnitude (Fig. 3) and additional contribution to the background surface ozone. The absolute magnitude is measured by "frequency \times duration \times ozone enhancement (in ppbv* hour)". It is commonly accepted that absolute magnitude peaks in spring, due to the springtime O₃ abundance in the lowermost stratosphere and frequent tropopause folding events. In this study, we found that a second peak in October over China because frequency, duration, ozone enhancement are all high in October. For the additional contribution to the overall surface ozone (Fig. 5), because the average background surface ozone concentrations in China are high in spring and summer (Fig. 5d). Therefore, these additional SITS ozone inputs would worsen the air pollution in spring and summer.

Additional references to consider are two papers which have studied stratospheric intrusions which impact surface ozone in the Hong Kong region:

Zhao et al. <https://doi.org/10.1016/j.atmosres.2020.105158>

Zhao, K., Huang, J., Wu, Y., Yuan, Z., Wang, Y., Li, Y., et al. (2021). Impact of stratospheric intrusions on ozone enhancement in the lower troposphere and implication to air quality in Hong Kong and other South China regions. *Journal of Geophysical Research: Atmospheres*, 126, e2020JD033955.

<https://doi.org/10.1029/2020JD033955>

And one that looks at dominant circulation patterns over parts of China

Yin, Z., Cao, B., and Wang, H.: Dominant patterns of summer ozone pollution in eastern China and associated atmospheric circulations, *Atmos. Chem. Phys.*, 19, 13933–13943, <https://doi.org/10.5194/acp-19-13933-2019>, 2019.

Response: Thank you very much for providing us these valuable references, which we have learnt a lot from, and also cited in this revision.

I would have liked to have seen the analysis include more links to the upper-air flow. The authors reference previous works (Line 347) but this requires the reader to go beyond the current manuscript to be convinced that the anomalous surface conditions are indeed connected to the stratosphere. Are there any other meteorological conditions with specific emission environments which could have led to this type of ozone-rich/CO-poor measurements? If the authors can prove this to me, I'd be more convinced that surface-measurements alone are sufficient. Otherwise, a case study showing how the method described on lines 265-270 works with the upper air dynamical features would go a long way.

Response: Thank you very much for these insightful comments and suggestions.

In this revision, we have made our best effort to further explain our SITS detecting methodology.

1. We explained our methodology in more details.

We explained that we incline to be conservative and set the detecting criteria rather strict. Therefore, the detected SITS events are highly likely to be the real cases.

Principally, stratospheric air parcels are characterized by high O₃ and low anthropogenic pollutant (such as CO), which can induce O₃ jump and CO reduction if they descend to the surface retaining their stratospheric features.

Using strict thresholds, we screen out those sharp and vigorous spikes in hourly concurrent O₃ and CO concentrations. These strict thresholds include (1) in a short period of time (1 hour), O₃ has to increase to 95th percentile level and CO has to decrease to its 5th percentile level at the same time, (2) at the beginning of SITS, O₃ concentrations have to exceed its seasonal mean value at the noontime, and (3) we assure that SITS would enhance surface O₃ concentrations, i.e. as long as surface O₃ concentrations are not above the background value, SITS stops. These thresholds are determined station by station and year by year, which are extreme enough to minimize the possibility of advection and recirculation of tropospheric air. In other words, only recent transport of stratospheric air can explain these sharp and vigorous variations at ground level.

2. This methodology is based on our previous studied on two cases, published in 'Atmospheric Chemistry and Physics' and 'npj Climate and Atmospheric Sciences'. The two cases were selected from the large samples of SITS, and were analyzed in detail concerning their stratospheric origins and impact on surface O₃. We applied multiple data sources including surface observations, radiosonde, reanalysis and trajectory simulations to investigate the surface O₃ and CO behaviors, vertical variations of RH and O₃, temporal evolution of PV and transport pathways of the airmass associated with the two SITS cases, all of which clearly indicated the stratospheric origins.

3. We added Extended Data Fig. 4. While it is not feasible to present detailed case studies of all SITS detected, we analyze the backward trajectories for all the 27K SITS events. Trajectories are initiated in surface stations when SITS events are detected, and the maximum height of each 10-day backward trajectory is regarded as the origins. The meteorological conditions (PV here) and O₃ concentrations along the trajectory are extracted from MERRA-2 reanalysis data. Given the different transport routes and travel time of trajectories of the 27K SITS events, the travel time was equally divided into three phases (referred to Stage 1, Stage 2 and Stage 3). Extended Data Fig. 4

shows a statistical overview of the height, PV, and O₃ concentrations associated with different stages of air parcels descending from high levels to the surface. We focus on the “Origin” stage. It shows that air parcels originate in the upper troposphere and lower stratosphere (UTLS) regions with a mean height of 250 hPa. With reference to the PV values and O₃ concentrations in the “Origin” stage, the air parcels reside above the dynamical tropopause (the 2-PVU iso-surface) where O₃ exceed 200 ppbv, showing prominent stratospheric features (e.g., Zhao et al., 2019; 2020). We have added these dynamical features of upper air in the revision as another validation of our detection results (Line 387-391).

Extended Data Fig. 4 Statistics of height, potential vorticity (PV) values, and O₃ concentrations during different travel stages of air mass associated with the SITS events in China over 2015-2022. The information of the start hour and location for each SITS is used to initiate HYSPLIT. The maximum height of each 10-day backward trajectory is regarded as the origins. PV values and O₃ concentrations along the trajectory are extracted from MERRA-2 reanalysis data. Given the different transport routes and travel time of trajectories of all the

SITS events, the travel time is evenly divided into three parts (referred to Stage 1, Stage 2 and Stage 3). The mean height, PV values and O₃ concentrations in each stage are shown by the red dots, and the blue arrows measure the 10th and 90th percentile of these parameters. The horizontal red line (2 PVU) in Extended Data Fig. 4b indicates the iso-surface of the dynamical tropopause.

In summary, we have confidence in the developed methodology because 1) we incline to be conservative and set the detecting criteria rather strict so the detected SITS events are highly likely to be the real cases, 2) the method is backed up by two published studies, 3) the detected SITS climatology compare well with literature in different locations, and 4) backward trajectory analyses for selected stations and all 27K trajectories suggest the stratospheric origins of the SITS events.

The manuscript is well written. I have the following minor and technical suggestions.

1. Line 11: change "to which" to "to what extent"

Response: Thanks to the reviewer for all careful corrections pointed out below. This is corrected here and throughout the text.

2. Line 55: The reference that this has become a highly debatable issue is from twenty years ago, and only one is provided. That does not seem very debatable if just one reference without any more recent debate.

Response: Thank you for pointing this out. We have removed the sentence as it is not strongly needed.

3. Line 85: change "locate" to "are located"

Response: Corrected.

4. Line 88: It would be helpful if elevation of these cities is quoted in the main text.

Response: Thank you for this suggestion. The elevation information is added (Line 91-92).

5. Line 94: More than reference #6 would be applicable here.

Response: Thanks. The original text was deleted in this version.

6. Line 116: It is at this stage I am wondering how these ozone enhancements have any connection to health impacts to motivate the study and the authors do not explain this till much later (line 153-162). I would encourage the authors to consider restructuring their message to keep the readers engaged in the motivation for the study.

Response: Good suggestion! In the revision, we rewrote the introduction and added the health impact of SITS (Line 36-45).

7. Line 142: Was a statistical test performed to determine that the enhancement was "significant"?

Response: Thanks for your point. We changed "significantly" to "substantially" in the revised manuscript.

8. Line 150: change "averagely" to "generally"

Response: Changed.

9. Line 157: change "sever" to "severe"

Response: Sorry for this typo. Corrected.

10. Line 158: Probably better to say over 70 % in warm months if considering March through August.

Response: Thanks for your suggestion. Revised in the manuscript. Also, the

month labels in the original Figure 5 were incorrect. In the revision, we corrected this mistake, and hence over 70% of SITS events are associated with O3 exceedances from the middle of spring to summer (Line 174).

11. Line 171: Provide the Figure reference to the end of the sentence.

Response: Added in the manuscript.

12. Line 179: write out stratospheric intrusion, in place of "SI".

Response: Thanks. Revised in the manuscript.

13. Line 204: Should Figure 6b also be referenced along with Supp. Figure 1b.

Response: Thank you for pointing this out. Added Figure 6b in the manuscript.

14. Line 205-207: Where are the numbers coming from quoted here? Is it off a figure? If it is coming from a supplemental figure, then the figure should be considered for the main text as substantive results should not be given from supplemental material.

Response: Thank you for pointing this out. The numbers come from Supplementary Figure 1 in the original version. Now we have moved this figure to the main text as Extended Data Fig. 2 in the revision.

15. Line 215-216: I do not follow the authors' connection here to the monthly accumulated ozone pollution hours during SITS in the supplemental figure 2. Likely the wrong figure is referenced here.

Response: Thanks for your careful review! Yes, the figure number is corrected as Supplement Figure 1.

16. Line 317: Add a space before "and"

Response: Thanks for your careful review. A space is added.

17. Line 401-402: MERRA-2 reanalysis data has DOIs per each file collection. These should be included in manuscripts.

Response: Sorry for this. We have added the websites where the data and model used in this study can be accessed in the revised manuscript.

18. References:

Danielsen, E. F., & Mohnen, V. A. (1977). Project dustorm report: Ozone transport, in situ measurements, and meteorological analyses of tropopause folding. *Journal of Geophysical Research*, 82(37), 5867–5877. <http://dx.doi.org/10.1029/JC082i037p05867>

Holton, J. R., Haynes, P. H., McIntyre, M. E., Douglass, A. R., Rood, R. B., & Pfister, L. (1995). Stratosphere-troposphere exchange. *Reviews of Geophysics*, 33(4), 403–439. <https://doi.org/10.1029/95RG02097>

Monks, P. S. (2000). A review of the observations and origins of the spring ozone maximum. *Atmospheric Environment*, 34(21), 3545–3561.

Response: Thank you so much for the provided references here and above to improve the quality of this research. We have cited these papers in this revision.

Reviewer #3 (Remarks to the Author):

Review of revised "Large stratospheric influence on surface ozone pollution in China" submitted to Nature Communications.

The authors addressed my comments. The manuscript is much improved and I recommend it for publication after a couple minor comments/technical edits are addressed as outlined below.

Line 28: remove "sometimes"

Line 39: change 'of' to 'to'

Line 45: remove "up". Simply "to date" is sufficient.

Line 69: My understanding of CO measurements is that they can be a bit jumpy by nature of the measurement precision of the instrument used. Something that is likely not necessary in the main text, but maybe in the added material, can you speak to the instrument precision/detection limits?

Line 73: move "the" from before advantage to before dense so it reads "Here, we take advantage of the dense surface observations..."

Line 78-79: Remove "Basing on large samples of SITS events detected across the nation in 8 years (27,616 events in total)," and begin this sentence with "In this work. In line 36, you mention SITS events are rare, and quoting 27,616 a) contradicts that statement and b) you explain these number in context in the following section nicely so this detail is not necessary in the introduction (I personally find it distracting).

Line 98-99: I found this sentence hard to follow. I suggest something like the following two options: "In central and eastern China, average surface O₃ enhancements are 15-25 ppbv during the SITS, while in western China, where elevations are high, only 7-15 ppbv." Or "In central and eastern China, average surface O₃ enhancements are 15-25 ppbv during the SITS, while only 7-15 ppbv in western China where elevations are high."

Line 104-106: This reads like possible future work with back trajectory analysis....but then I read further on and you've done back-trajectory analysis in the additional material. Can you connect this "Possibly, SITS events over plain regions in eastern China may initiate at higher altitudes within the stratosphere" to work you've already done?

Line 143: Can this notation be clearer? Is it ppbv per hour? The * does not make sense to me.

Line 171-172: It is unclear in this new text that Figure 5 is considering all three O₃ threshold standards. I recommend swapping the order of these sentences and adding this detail, something like: "Figure 5 shows the fraction of SITS events with O₃ exceedances (based on

the three O3 thresholds) to the total number of SITS events in each month, averaged over all the stations and all the years. Here, for each standard, if there is at least one O3 exceedance during a SITS event, we regard the event as a SITS-induced O3 exceedance.”

Line 189: I know there was at least one place I suggested changing SI to “stratospheric intrusions” in the first version, but now SI is defined at the start of the manuscript. This definition should be used throughout if it is defined.

Line 193: The zero hour has a weird sharp transition in the SITS O3 such that the curve looks more like Non_SITS O3 after the zero hour. Is this discussed? Does this have to do with the time of day of SITS events occurring?

Line 387-408: These results on back-trajectory are discussed before the back-trajectory has been introduced in the following section 409. Some careful restructuring should be done.

Line 414-415: As mentioned in my first review, MERRA-2 reanalysis data has DOIs per each dataset and these should be included in the manuscripts.
<https://disc.gsfc.nasa.gov/datasets?project=MERRA-2> see this link to find the datasets used and their documentation. In addition, Gelaro et al. 2017 (reference number 64) should be referenced when MERRA-2 is defined on line 415, not just after the link on line 416.

Review by K. Emma Knowland

Review of revised “Large stratospheric influence on surface ozone pollution in China” submitted to Nature Communications.

The authors addressed my comments. The manuscript is much improved and I recommend it for publication after a couple minor comments/technical edits are addressed as outlined below.

Thank you for your suggestions and comments that improve the quality of this paper.

Line 28: remove “sometimes”

Changed.

Line 39: change ‘of’ to ‘to’

Changed.

Line 45: remove “up”. Simply “to date” is sufficient.

Changed.

Line 69: My understanding of CO measurements is that they can be a bit jumpy by nature of the measurement precision of the instrument used. Something that is likely not necessary in the main text, but maybe in the added material, can you speak to the instrument precision/detection limits?

Thanks for this suggestion. The detection limits and precision of ozone and CO analyzed are added in the Methods section Line 383-385.

Line 73: move “the” from before advantage to before dense so it reads “Here, we take advantage of the dense surface observations...”

Changed.

Line 78-79: Remove “Basing on large samples of SITS events detected across the nation in 8 years (27,616 events in total),” and begin this sentence with “In this work. In line 36, you mention SITS events are rare, and quoting 27,616 a) contradicts that statement and b) you explain these number in context in the following section nicely so this detail is not necessary in the introduction (I personally find it distracting).

Thanks for your points. We have added a condition for the statement in line 36, “SITS events are transient and limited in local areas”, In the current sentence, we also added a condition “SITS events detected locally at individual stations”. Both statements illustrate that SITS events are transient and limited in local areas. Because SITS can impact large areas, the total number of the locally detected SITS is large, which ensure the robustness of this study.

Line 98-99: I found this sentence hard to follow. I suggest something like the following two options:

“In central and eastern China, average surface O₃ enhancements are 15-25 ppbv during the SITS, while in western China, where elevations are high, only 7-15 ppbv.” Or “In central and eastern China, average surface O₃ enhancements are 15-25 ppbv during the SITS, while only 7-15 ppbv in western China where elevations are high.”

Thanks. We changed the expression accordingly.

Line 104-106: This reads like possible future work with back trajectory analysis...but then I read further on and you've done back-trajectory analysis in the additional material. Can you connect this “Possibly, SITS events over plain regions in eastern China may initiate at higher altitudes within the stratosphere” to work you've already done?

Thanks for pointing this out. Recently we are performing analysis of trajectory associated with SITS, and the preliminary results show that SITS events reaching plain regions have a deep origin in the stratosphere. We will conduct more tests and analysis of the temporal and spatial variations in SITS origins in the following work.

Line 143: Can this notation be clearer? Is it ppbv per hour? The * does not make sense to me.

The amount of injected stratospheric O₃ (O_3^{strat}) is calculated by integrating the excess of O₃ concentrations above their reference values based on hourly O₃ measurements (Equation 1 in Methods section). To be more specific, the difference between O₃ and the corresponding baseline are summed up in each hour during the SITS periods. So, the units of O_3^{strat} comes to ppbv*hour.

Line 171-172: It is unclear in this new text that Figure 5 is considering all three O₃ threshold standards. I recommend swapping the order of these sentences and adding this detail, something like: “Figure 5 shows the fraction of SITS events with O₃ exceedances (based on the three O₃ thresholds) to the total number of SITS events in each month, averaged over all the stations and all the years. Here, for each standard, if there is at least one O₃ exceedance during a SITS event, we regard the event as a SITS-induced O₃ exceedance.”

Thank you for your suggestions. We have modified the sentences accordingly.

Line 189: I know there was at least one place I suggested changing SI to “stratospheric intrusions” in the first version, but now SI is defined at the start of the manuscript. This definition should be used throughout if it is defined.

Thanks. We have checked and modified the definition throughout the manuscript.

Line 193: The zero hour has a weird sharp transition in the SITS O₃ such that the curve looks more

like Non_SITS O₃ after the zero hour. Is this discussed? Does this have to do with the time of day of SITS events occurring?

In the SITS detection method, both the absolute magnitudes and hourly variations of O₃ and CO must exceed their thresholds in order to determine the exact hour (i.e., the zero hour) when SITS starts. As a result, the most pronounced spikes appear in the zero hour leading to the sharp transition. We agree that the curve of SITS O₃, which is the mean of a large ensemble, can be influenced by the time of day of SITS events occurring. We will investigate the preferred time and synoptic conditions for SITS. As mentioned above, there are many issues, such as the origin, temporal and spatial distribution, meteorological triggers of SITS events, remain to be addressed in future work.

Line 387-408: These results on back-trajectory are discussed before the back-trajectory has been introduced in the following section 409. Some careful restructuring should be done.

Thank you for pointing this out. We have added “see details of the backward trajectory simulations in the following section” in Line 360.

Line 414-415: As mentioned in my first review, MERRA-2 reanalysis data has DOIs per each dataset and these should be included in the manuscripts. <https://disc.gsfc.nasa.gov/datasets?project=MERRA-2> see this link to find the datasets used and their documentation. In addition, Gelaro et al. 2017 (reference number 64) should be referenced when MERRA-2 is defined on line 415, not just after the link on line 416.

Thanks. We have provided the DOI for the MERRA-2 dataset I and changed the location of MERRA-2 reference.

Review by K. Emma Knowland